# Transporter-mediated depletion of extracellular proline directly contributes to plant pattern-triggered immunity against a bacterial pathogen

Conner J. Rogan[1], Yin-Yuin Pang[1], Sophie D. Mathews[1], Sydney E. Turner[1], Alexandra J. Weisberg [1], Silke Lehmann [2], Doris Rentsch [2] & Jeffrey C. Anderson [1] ✉

Plants possess cell surface-localized immune receptors that detect microbe-associated molecular patterns (MAMPs) and initiate defenses that provide effective resistance against microbial pathogens. Many MAMP-induced signaling pathways and cellular responses are known, yet how pattern-triggered immunity (PTI) limits pathogen growth in plants is poorly understood. Through a combined metabolomics and genetics approach, we discovered that plant-exuded proline is a virulence-inducing signal and nutrient for the bacterial pathogen *Pseudomonas syringae*, and that MAMP-induced depletion of proline from the extracellular spaces of Arabidopsis leaves directly contributes to PTI against *P. syringae*. We further show that MAMP-induced depletion of extracellular proline requires the amino acid transporter Lysine Histidine Transporter 1 (LHT1). This study demonstrates that depletion of a single extracellular metabolite is an effective component of plant induced immunity. Given the important role for amino acids as nutrients for microbial growth, their depletion at sites of infection may be a broadly effective means for defense against many pathogens.

All organisms must detect and defend against pathogens. In both plants and animals, pattern recognition receptors (PRRs) protect against infection by detecting conserved microbial features termed microbe or pathogen associated molecular patterns (M/PAMPs)[1,2]. In animals, recognition of MAMPs by PRRs activates specialized mobile immune cells that coordinate their functions with local tissues to kill and clear pathogens from infection sites[3]. In contrast, plants do not possess mobile immune cells that seek out and destroy invading pathogens. Rather, individual cells that constitute plant tissues are capable of directly detecting and mitigating threats locally at sites of infection[4]. Because many plant pathogens remain outside of host cells during infection, PRR-mediated resistance must be capable of

hindering microbial growth within the apoplast, or extracellular space, of plant tissues. In this regard, activation of PRRs triggers apoplast-localized defense responses, including the production of reactive oxygen species, secretion of antimicrobial compounds, and cell wall reinforcement[5]. Collectively, MAMP-induced defenses provide effective resistance termed MAMP- or pattern-triggered immunity (PTI)[4]. However, the exact mechanism(s) of how PTI limits pathogen infection of the apoplast remain(s) largely unknown.

*Pseudomonas syringae* are Gram-negative bacteria that cause disease on economically important crops as well as the model plant Arabidopsis[6]. *P. syringae* are primarily foliar pathogens and enter leaves through openings such as stomata or wounds. Once inside,

[1]Department of Botany and Plant Pathology, Oregon State University, Corvallis, OR 97331, USA. [2]Institute of Plant Sciences, University of Bern, Bern, Switzerland. ✉e-mail: anderje2@oregonstate.edu

*P. syringae* can multiply to high levels within the leaf apoplast. A key virulence factor for *P. syringae* is its type III secretion system (T3SS), a syringe-like apparatus that delivers PTI-suppressing effector proteins into plant cells[6]. Although the T3SS is critical for virulence, T3SS-encoding genes are not constitutively expressed and must be induced during infection. To this end, *P. syringae* relies on plant-exuded metabolites, namely sugars and specific amino acids, as signals to induce its T3SS-encoding genes[7]. *P. syringae* receptors and signaling pathways required for detecting specific host signals are known, and mutants lacking these signaling components are less virulent[7–9]. On the host side, the extracellular release of T3SS-inducing metabolites by plant cells is genetically-regulated, as evidenced by the discovery of an Arabidopsis mutant lacking *MAP KINASE PHOSPHATASE1* (*MKP1*) that exudes lower amounts of several T3SS-inducing metabolites and as a consequence is more resistant to *P. syringae* infection[10,11]. Collectively, these observations reveal that the abundance of virulence-inducing metabolites encountered by *P. syringae* in plant tissues is an important determinant of infection outcomes.

Pathogenic bacteria must express specific virulence genes to infect a host. As such, the signaling events that initiate the expression of these genes represent a vulnerability that can be targeted by host defenses[7]. A long-standing observation is that activation of plant immunity can prevent *P. syringae* from injecting type III effectors into plant cells. In this regard, *P. syringae* is restricted in its ability to inject effectors into Arabidopsis and tobacco leaf cells pre-treated with MAMPs[12]. A similar restriction occurs when *P. syringae* infects MAMP-treated Arabidopsis seedlings[10]. More recent transcriptome analyses revealed that expression of T3SS-encoding genes is inhibited during *P. syringae* infection of MAMP-treated Arabidopsis leaves, suggesting that PTI limits *P. syringae* growth by interfering with T3SS gene induction[13,14]. In this work we investigated MAMP-induced changes in the metabolic composition of the Arabidopsis leaf apoplast. Our goal was to determine whether any observed changes in metabolite levels within the apoplast may be causal for the MAMP-induced restriction in T3SS deployment by *P. syringae*.

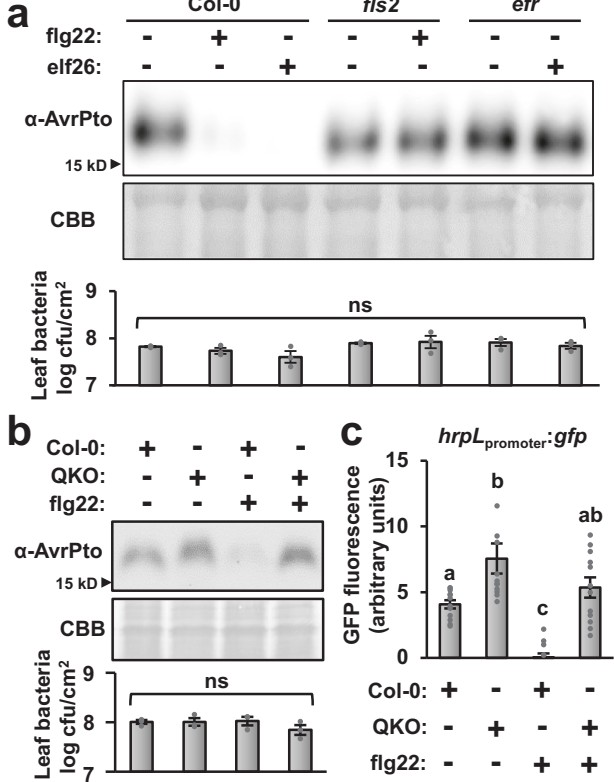

**Fig. 1 | Activation of pattern-triggered immunity in Arabidopsis leaves prevents *P. syringae* from expressing type III secretion system-encoding genes. a** Leaves of four-week-old Arabidopsis wild-type Col-0, *fls2* and *efr* plants were syringe-infiltrated with 100 nM flg22, 100 nM elf26, or a mock treatment. After five hours, the treated leaves were infiltrated with 5 ×10⁸ cfu/mL of wild-type *P. syringae* DC3000. The upper panel is an anti-AvrPto immunoblot of total protein extracts from infected leaf tissue, middle panel is Coomassie Blue (CBB) staining of the immunoblot to assess equal loading, and lower panel is a graph of bacteria levels in the infected tissue samples. Graphed in the lower panel are means ± SE of log-transformed counts of colony forming units (cfu). **b** Leaves of four-week-old Arabidopsis wild-type Col-0 and *sid2/pad4/ein2/dde2* (QKO) plants were syringe-infiltrated with 100 nM flg22 or a mock treatment. After five hours, the treated leaves were infiltrated with 5×10⁸ cfu/mL of wild-type *P. syringae* DC3000. The upper panel is an anti-AvrPto immunoblot of total protein extracts from infected leaf tissue, middle panel is Coomassie Blue (CBB) staining of the immunoblot to assess equal loading, and lower panel is a graph of bacteria levels in the infected tissue samples. Graphed in the lower panel are means ± SE of log-transformed counts of colony forming units (cfu). **c** Leaves of four-week-old Arabidopsis wild-type Col-0 and QKO plants were syringe-infiltrated with 100 nM flg22 or a mock treatment. After five hours, the treated leaves were infiltrated with 5×10⁸ cfu/mL of *P. syringae* DC3000 carrying *hrpL*promoter:*gfp*, or a DC3000 empty vector control strain. GFP fluorescence from leaf tissue six hours post-infiltration with DC3000 *hrpL*promoter:*gfp*. Graphed are means ± SE of GFP fluorescence normalized to fluorescence from tissue infiltrated with a DC3000 empty vector control strain, *n* = 12. Lower case letters in all panels denote significance groupings based on ANOVA with Tukey's HSD, *p* < 0.05. The abbreviation ns is not significant. Data in all panels are representative of three independent experiments.

## Results

### Pattern-triggered immunity restricts expression of *P. syringae* type III secretion-encoding genes

We first verified that PTI restricts the expression of T3SS-encoding genes in *P. syringae* under our experimental conditions. We syringe-infiltrated *P. syringae* pathovar *tomato* strain DC3000 (herein DC3000) into Arabidopsis wild-type Col-0 leaves pre-treated for five hours with the MAMP flg22 or a mock treatment, then used immunoblotting to detect the levels of type III effector AvrPto in the infected tissue. AvrPto levels were greatly diminished in flg22-treated leaves infected with DC3000 for five hours (Fig. 1a). The reduced amount of AvrPto was not due to differences in leaf bacteria populations at this early time point (Fig. 1a). We observed identical results with leaves treated with the MAMP elf26, indicating that decreased AvrPto accumulation occurs in response to multiple MAMPs (Fig. 1a). Flg22 and elf26 are recognized by the PRRs FLS2 and EFR, respectively[15,16]. AvrPto levels were not altered in *fls2* or *efr* mutants pretreated with either flg22 or elf26, respectively, confirming the observed phenotype is receptor-dependent (Fig. 1a).

Plant defense responses including PTI are severely compromised in an Arabidopsis *dde2 ein2 pad4 sid2* quadruple knockout (QKO) mutant deficient in the production of defense hormones salicylic acid, ethylene and jasmonic acid[17]. In contrast to reduced AvrPto accumulation in flg22-treated Col-0 leaves, no decrease in AvrPto abundance was detected in flg22-treated QKO leaves infected with DC3000 (Fig. 1b). We also infiltrated Col-0 and QKO leaves with a DC3000 *hrpL*promoter:*gfp* reporter strain and, based on GFP fluorescence from the infected tissue, observed that expression of T3SS regulator *hrpL* was significantly decreased in flg22-treated Col-0 leaves, whereas a significant flg22-dependent decrease in *hrpL* expression did not occur in QKO leaves (Fig. 1c). Indeed, *hrpL* expression was significantly higher in mock-treated QKO compared to mock-treated Col-0, possibly due to the attenuated response of QKO to endogenous DC3000 MAMPs (Fig. 1c). Together, these results confirm that flg22-and elf26-induced defenses restrict the expression of T3SS genes in *P. syringae*, and reveal that flg22-induced restriction of T3SS genes does not occur in QKO mutant leaves.

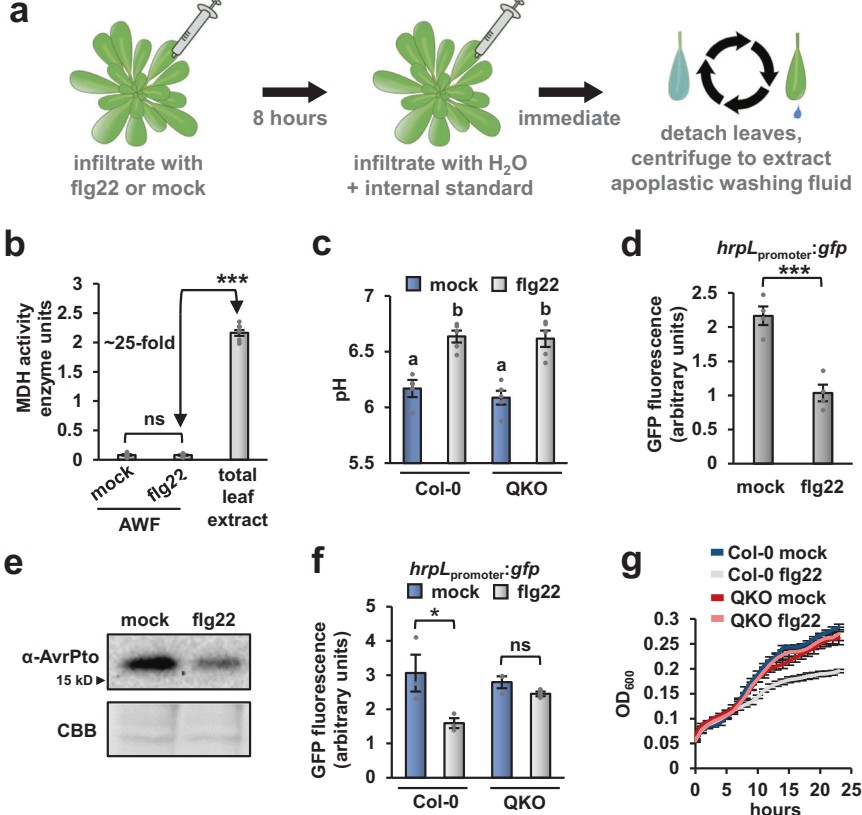

**Fig. 2 | Apoplastic wash fluid isolated from flg22-treated Arabidopsis leaves inhibits *P. syringae* type III secretion and growth.** Leaves of four-week-old Arabidopsis were syringe-infiltrated with 100 nM flg22 or a mock solution. After eight hours, apoplastic wash fluid (AWF) was isolated from the same leaves. **a** Diagram of AWF extraction procedure. **b** Measurements of malate dehydrogenase (MDH) activity in AWF and total leaf extracts from wild-type Col-0 leaves. Graphed are means ± SE of MDH enzyme units, *n* = 8. Data are from eight independent experiments. **c** The pH of AWF from mock- and flg22-treated wild-type Col-0 and *sid2/pad4/ein2/dde2* (QKO) mutant leaves immediately after extraction. Graphed are means of AWF pH ± SE, *n* = 5. Data are from five independent experiments. Lower case letters denote significance groupings based upon ANOVA with Tukey's HSD, *p* < 0.01. **d** *P. syringae* DC3000 carrying a *hrpL_promoter*:*gfp* or empty vector control plasmid were cultured for 10 h in AWF from Col-0 leaves treated with either flg22 or a mock treatment. Graphed are

means ± SE of normalized GFP fluorescence, *n* = 4. **e** AvrPto abundance in DC3000 incubated for six hours in AWF from mock- or flg22-treated Col-0 leaves. Upper panel is chemiluminescent signal from anti-AvrPto immunoblot of total protein extracts and lower panel is Coomassie Brilliant Blue (CBB) staining to assess equal loading. **f** *P. syringae* DC3000 carrying a *hrpL_promoter*:*gfp* or empty vector control plasmid were incubated for 10 h in AWF from wild type Col-0 and QKO leaves treated with either flg22 or a mock treatment. Graphed are means ± SE of normalized GFP fluorescence, *n* = 3. **g** Growth of *P. syringae* DC3000 *hrpL_promoter*:*gfp* in AWF from Col-0 and QKO leaves treated with either flg22 or a mock treatment. Graphed are means of culture optical density at λ = 600 nm (OD600) measurements ± SE, *n* = 3. Data in panels **d**–**g** are representative of results from three independent experiments. Asterisks in panels **b**, **d** and **f** denote statistical significance based on two-sided *t*-test. *\*p* < 0.05, *\*\*p* < 0.01, *\*\*\*p* < 0.001, ns is *p* > 0.05.

## Apoplastic wash fluid from MAMP-treated leaves inhibits *P. syringae* growth and T3SS deployment

To investigate how PTI restricts T3SS deployment, we isolated apoplastic wash fluid (AWF) from mock- and MAMP-treated Arabidopsis leaves. For this procedure, we syringe-infiltrated leaves with water containing flg22 or the solvent DMSO as a mock control. After eight hours, we then syringe-infiltrated the same leaves with water and used low-speed centrifugation to isolate AWF from the infiltrated leaves (Fig. 2a). Using a malate dehydrogenase (MDH) enzyme assay to detect cytoplasmic proteins, we confirmed that our AWF isolation procedure did not result in substantial cytoplasmic leakage (Fig. 2b). Furthermore, no significant difference in MDH activity between AWF from mock-treated (mock-AWF) or flg22-treated (flg22-AWF) leaves was detected (Fig. 2b).

The leaf apoplast is rapidly alkalinized following MAMP perception[18,19]. Consistent with these previous observations, the average pH of our mock-AWF samples was 6.17, whereas the average pH of flg22-AWF samples was 6.63 (Fig. 2c). We measured a similar increase in pH between mock- and flg22-AWF from PTI-deficient QKO leaves (Fig. 2c). Based on these data, alkalization of the apoplast is likely not causal for flg22-induced restriction of T3SS genes.

We next tested whether T3SS deployment is altered when *P. syringae* are cultured in AWF from flg22-treated Col-0 leaves. To alleviate possible pH effects, we mixed the extracted AWF with a phosphate-buffered minimal medium (MM). The added MM was sufficient to adjust the pH of both mock-AWF and flg22-AWF to 6.0 (Fig. S1a). We then cultured DC3000 *hrpL_promoter*:*gfp* in the buffered AWF and, based on GFP fluorescence, observed significantly decreased *hrpL* expression in bacteria cultured in flg22-AWF compared to mock-AWF (Fig. 2d). To further investigate this difference in AWF bioactivity, we chloroform-extracted mock-AWF and flg22-AWF, lyophilized the resulting aqueous and organic phases to dryness, then resuspended the dried samples directly into MM. We cultured DC3000 *hrpL_promoter*:*gfp* in these extracted samples and observed that nearly all of the *hrpL*-inducing activity was present in the aqueous fraction (Fig. S1b). Furthermore, decreased *hrpL*-inducing activity of flg22-AWF relative to mock-AWF was evident in the aqueous but not in the organic fractions (Fig. S1b). Immunoblot detection of AvrPto in the treated bacteria confirmed the difference in bioactivity of aqueous mock- and flg22-AWF fractions (Fig. 2e). We also incubated DC3000 *hrpL_promoter*:*gfp* in mock-AWF and flg22-AWF isolated from QKO leaves. In contrast to AWF from Col-0 leaves, no

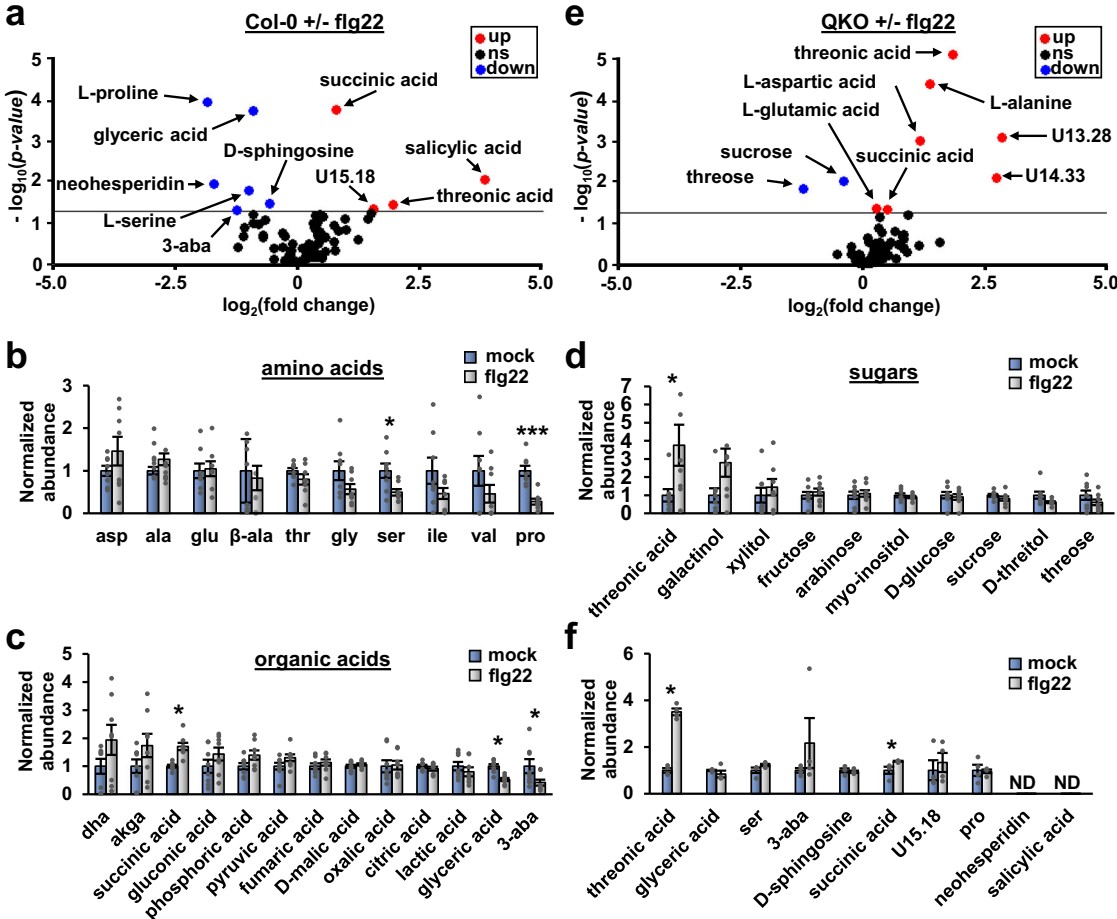

**Fig. 3 | Metabolomics analyses of apoplastic wash fluid isolated from flg22-treated Arabidopsis leaves.** Leaves of four-week-old Arabidopsis plants were syringe-infiltrated with 100 nM flg22 or a mock solution. After 8 h, apoplastic wash fluid (AWF) was isolated from the treated leaves and analyzed by GC-MS. **a** Volcano plot of average fold change values and associated *p* values for compounds detected in AWF extracted from Col-0 leaves treated with flg22 or a mock treatment, *n* = 8. Data were pooled from eight independent experiments. The horizontal line denotes *p* = 0.05 based on two-sided *t*-test. Compounds that significantly increased or decreased in AWF from flg22-treated leaves are represented in red and blue, respectively. Abbreviation 3-aba is 3-aminoisobutyric acid. **b-d** Relative abundances of (**b**) amino acids, (**c**) organic acids or (**d**) sugars detected in AWF from mock- and flg22-treated Col-0 leaves. Abundance values were normalized to a value of 1 in mock samples. Graphed are means ± SE of metabolite abundance, *n* = 8.

Abbreviations are dha, dehydroxyascorbic acid; akga, alpha ketoglutaric acid; 3-aba, 3-aminoisobutyric acid; β-ala, β-alanine. Standard three letter abbreviations are used for proteinaceous amino acids. Data are the same as in panel **a**. **e** Volcano plot of average fold change and associated *p* values for compounds detected in AWF extracted from *sid2/pad4/ein2/dde2* (QKO) leaves treated with flg22 or a mock treatment, *n* = 4. Plot labels are the same as for panel **a**. Data were pooled from four independent experiments. The horizontal line denotes *p* = 0.05 based on two-sided *t*-test. **f** Average relative abundances of metabolites in mock and flg22-treated QKO AWF that were significantly altered in abundance between mock- and flg22-treated Col-0 AWF. Graphed are means ± SE of metabolite abundance, *n* = 4. Data are the same as in panel **e**. ND is not detected. Asterisks in panels **b**−**f** denote statistical significance based on two-sided *t*-tests. *p* < 0.05, ***p* < 0.001.

significant decrease in *hrpL*-inducing activity was detected between mock-AWF and flg22-AWF isolated from QKO leaves (Fig. 2f).

In addition to T3SS induction, we also tested for DC3000 growth in AWF. Similar to the pattern for *hrpL* expression, DC3000 grew in mock-AWF from Col-0 leaves, and this growth was significantly reduced in flg22-AWF from Col-0 leaves (Fig. 2g). After chloroform extraction of AWF samples, the growth-stimulating activity in AWF was present in only the aqueous fraction (Fig. S1c, d). No difference in bacterial growth was observed for DC3000 cultured in mock- and flg22-AWF from QKO plants (Fig. 2g). Based on these results, we conclude that the aqueous phase of AWF isolated from flg22-treated Col-0 leaves has decreased T3SS- and bacterial growth-inducing activity.

### Metabolomics analysis of apoplastic wash fluid from flg22-treated Arabidopsis leaves

To determine the molecular basis for the decreased T3SS-inducing activity of flg22-AWF, we used gas chromatography-mass spectrometry (GC-MS) to profile flg22-induced changes in the Arabidopsis leaf apoplast metabolome. From samples collected in eight independent experiments, we identified a total of 96 features from both mock- and flg22-AWF, 62 of which could be identified based on comparisons of peak retention times and mass spectra to entries in the FiehnLib library[20]. Based on *p* < 0.05 and >2 fold-change cutoffs, four features increased and six decreased in flg22-AWF compared to mock-AWF (Fig. 3a). Notably, the defense hormone salicylic acid increased 13-fold in flg22-AWF, thus confirming that defense responses were elicited in flg22-treated leaves[21]. We grouped the identified metabolites into categories of amino acids (Fig. 3b), organic acids (Fig. 3c) and sugars (Fig. 3d). Despite surprisingly few changes overall, significant differences in metabolite abundance between mock- and flg22-AWF occurred in each of these three categories. Notably, threonic acid was the only sugar that differentially accumulated between treatments, increasing four-fold in flg22-AWF. We also profiled metabolites present in mock-AWF and flg22-AWF collected from QKO leaves (Fig. 3e). Of those metabolites that

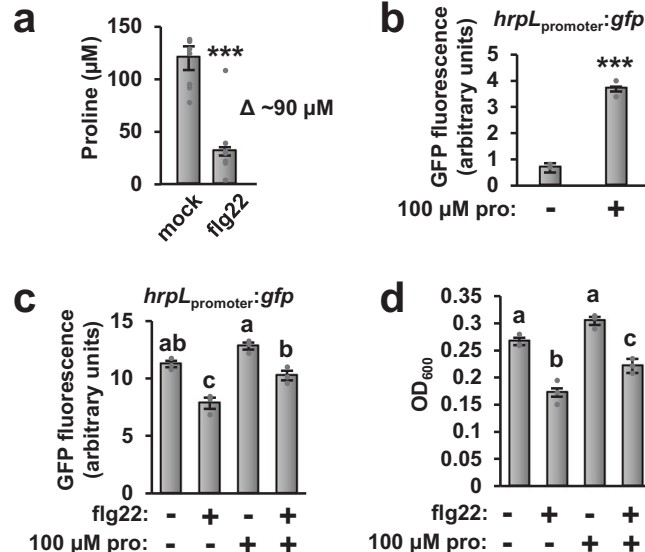

**Fig. 4 | Decreased proline levels contribute to decreased T3SS- and growth-inducing activity of apoplastic wash fluid from flg22-treated Arabidopsis leaves. a** Absolute quantification of proline levels in apoplastic wash fluid (AWF) from mock-and flg22-treated leaves. Graphed are means ± SE of proline abundance, $n = 8$. Data were pooled from eight independent experiments. **b**, **c** *P. syringae* DC3000 *hrpL*~promoter~:*gfp* were cultured in (**b**) minimal medium containing 10 mM fructose and supplemented with or without 100 μM proline, or (**c**) in mock- and flg22-AWF supplemented with or without 100 μM proline. Graphed are means ± SE of normalized GFP fluorescence, $n = 3$. **d** Growth of *P. syringae* DC3000 *hrpL*~promoter~:*gfp* in mock- and flg22-AWF isolated from Col-0 leaves and supplemented with or without 100 μM proline. Graphed are means of culture optical density at λ = 600 nm ± SE, $n = 3$. Data in **b**–**d** are representative of three independent experiments. Asterisks in panels **a** and **b** denote significance based on two-sided *t*-test, \*\*\**p* < 0.001. Lower case letters in **c** and **d** denote significance groupings based on ANOVA with Tukey's HSD, *p* < 0.05.

significantly changed in abundance in flg22-AWF from Col-0 leaves, only threonic acid and succinic acid also significantly accumulated in flg22-AWF from QKO (Fig. 3f), indicating that flg22-induced accumulation of these two metabolites is not causal for decreased T3SS-inducing bioactivity of flg22-AWF.

We next measured the absolute concentrations of metabolites that differentially accumulated in AWF in response to flg22 treatment. In mock-AWF, all of these metabolites were detected at concentrations <20 μM except for proline and serine that were present at 120 μM and 194 μM, respectively (Fig. 4a, S2a, b). We then tested each of these metabolites for *hrpL*-inducing activity at their measured concentrations. Only proline and serine significantly induced *hrpL* expression (Fig. 4b and S2c). Based on the abundance and T3SS-inducing activity of proline and serine, we hypothesized that decreased abundance of these specific amino acids is causal for the decreased T3SS-inducing bioactivity of flg22-AWF. We focused on proline because, compared to serine, proline was a more potent inducer of *hrpL* expression (Fig. S2c) and showed the greatest flg22-dependent fold-change of all metabolites except for salicylic acid (Fig. 3a). We added 100 μM of proline, a similar concentration to the ~90 μM difference in proline between mock- and flg22-AWF (Fig. 4a), to both mock- and flg22-AWF. The added proline restored *hrpL* expression in flg22-AWF back to levels observed with mock-AWF (Fig. 4c). Moreover, adding proline to the flg22-AWF partially yet significantly restored DC3000 growth to levels measured in mock-AWF (Fig. 4d). Together these results revealed that decreased proline contributed to the reduced T3SS- and growth-inducing activity of flg22-AWF.

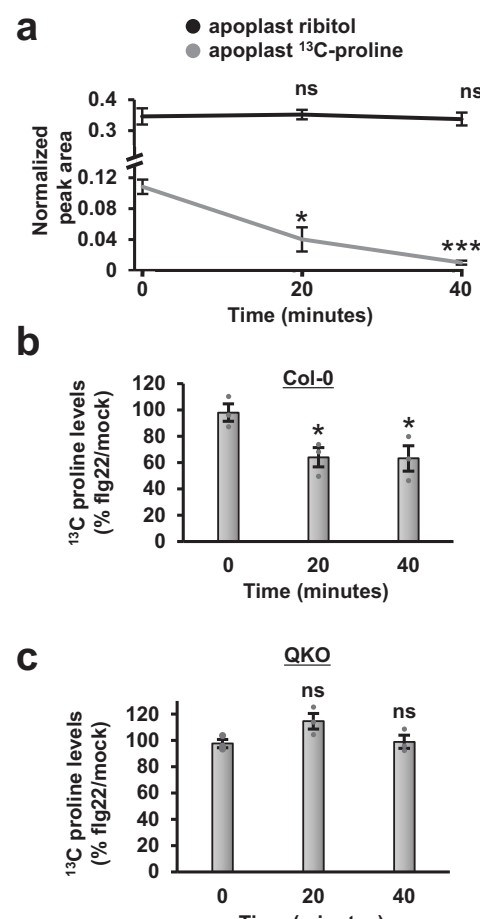

**Fig. 5 | Flg22 stimulates removal of proline from the Arabidopsis leaf apoplast. a** Arabidopsis Col-0 leaves were infiltrated with 500 μM $^{13}$C-proline and 164 μM ribitol. Apoplastic wash fluid (AWF) was isolated from treated leaves at time points indicated, and the abundance of $^{13}$C-proline and ribitol in AWF measured by GC-MS. Graphed are means ± SE of $^{13}$C-proline or ribitol peak area normalized to levels of myristic acid added as an external standard, $n = 3$. **b**, **c** Col-0 (**b**) and *sid2/pad4/ein2/pad4* (QKO) (**c**) plants were treated with 100 nM flg22 or a mock treatment for six hours. The plants were then infiltrated with 500 μM $^{13}$C-proline and 164 μM ribitol. AWF was recovered from treated leaves at time points indicated. Graphed are means ± SE of $^{13}$C-proline peak area normalized to ribitol peak area, $n = 3$. Asterisks in all panels denote significance based on two-sided *t*-test vs t = 0 measurements, \**p* < 0.05, \*\*\**p* < 0.001, ns is *p* > 0.05. Data in each panel were pooled from three independent experiments.

## Flg22 stimulates an increased rate of proline removal from the leaf apoplast

We next examined whether flg22 treatment alters the rate of proline depletion from the apoplast of Arabidopsis leaves. As a first step, we infiltrated a solution of 500 μM $^{13}$C-labelled proline and 164 μM ribitol as an internal standard into naïve Col-0 leaves, and used GC-MS to measure the abundance of $^{13}$C-proline and ribitol in AWF isolated from the infiltrated leaves. $^{13}$C-proline and endogenous $^{12}$C-proline were clearly differentiated based on shifts in mass spectra peaks (Fig. S3a). $^{13}$C-proline levels in AWF rapidly decreased over a 40 min time course, whereas levels of ribitol remained unchanged (Fig. 5a). A similar rate of $^{13}$C-proline depletion was observed in AWF from leaves detached from the rosette, thus ruling out long distance vascular transport contributing to $^{13}$C-proline depletion, though these data do not rule out local uptake of $^{13}$C-proline into vascular tissues (Fig. S3b). We then examined the rate of $^{13}$C-proline depletion from the apoplast of Col-0 and QKO leaves pre-treated with flg22 for six hours. In comparison to mock-AWF from Col-0 leaves, the levels of $^{13}$C- proline in flg22-AWF

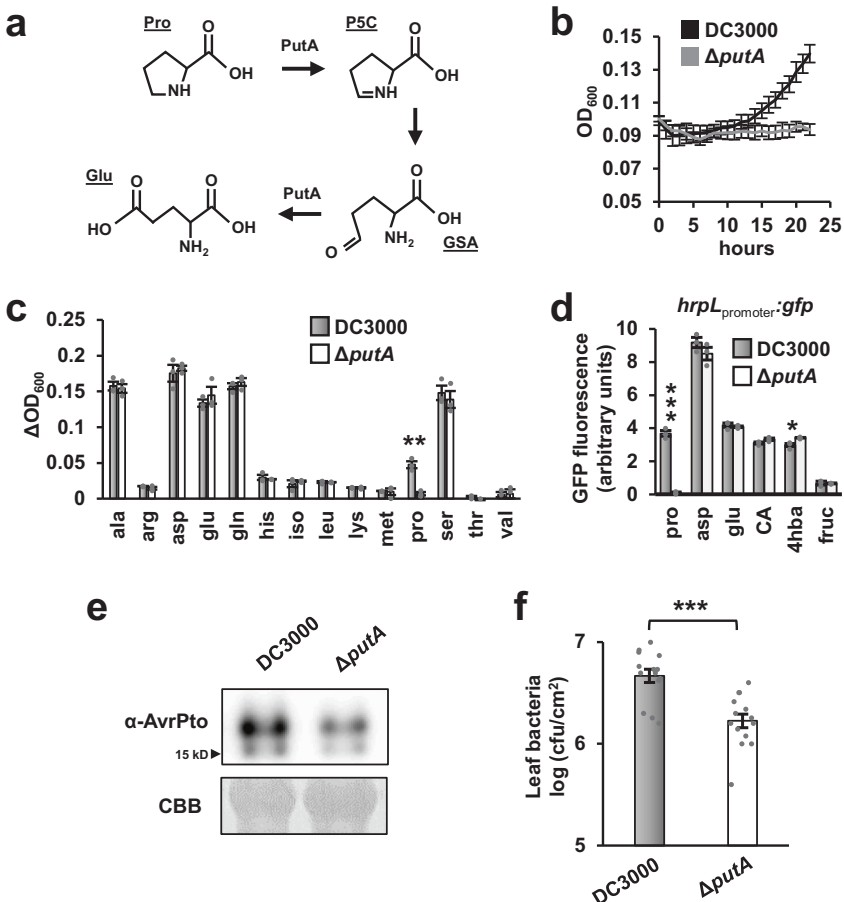

**Fig. 6 | *P. syringae* proline utilization A (*putA*) is required for maximal type III secretion system deployment and bacterial growth during infection of Arabidopsis. a** Reactions catalyzed by PutA. Pro=proline, P5C = $\Delta^1$-pyrroline-5-carboxylate, GSA = glutamic semialdehyde, Glu=glutamic acid. **b** Time course of *P. syringae* DC3000 and DC3000Δ*putA* growth in M9 medium supplemented with 10 mM proline. Graphed are means ± SE of optical density at $\lambda = 600$ nm ($OD_{600}$) measurements at indicated time points, $n = 3$. **c** Growth of *P. syringae* DC3000 and DC3000Δ*putA* in M9 medium supplemented with 10 mM of individual amino acids as indicated. Graphed are means ± SE of $OD_{600}$ measurements at 22 h post-inoculation relative to $OD_{600}$ at $t = 0$, $n = 3$. **d** GFP fluorescence from *P. syringae* DC3000 *hrpL_promoter:gfp* and DC3000Δ*putA hrpL_promoter:gfp* reporter strains incubated for 4.5 h in minimal medium supplemented with 10 mM fructose and 200 μM of individual metabolites as indicated. CA = citric acid, 4hba = 4-hydroxybenzoic acid, fruc = fructose. Graphed are means ± SE of

normalized GFP fluorescence, $n = 3$. **e** AvrPto protein abundance in leaf tissue five hours after infiltration of leaves with either *P. syringae* DC3000 or DC3000Δ*putA*. Upper panel is immunoblot detection of AvrPto levels in protein extracts from infected leaves, lower panel is Coomassie Brilliant Blue (CBB) staining of the immunoblot to assess equal loading. Data shown are representative of three independent experiments. **f** Leaf bacteria levels three days after infiltration of Arabidopsis leaves with $1 \times 10^6$ cfu/mL of *P. syringae* DC3000 or DC3000Δ*putA*. Graphed are log transformed means ± SE of colony-forming units (cfu) of bacteria isolated from infected tissue, $n = 14$. Data were pooled from three independent experiments, $n = 4$-5 per experiment. Results in panels **b**–**d** are representative of two independent experiments. Asterisks in panels **c**, **d** and **f** denote statistical significance based on two-sided *t*-test, *$p < 0.05$. **$p < 0.01$, ***$p < 0.001$.

from Col-0 leaves were significantly reduced by more than 35% at both 20 and 40 min post-infiltration (Fig. 5b). In comparison, no significant difference in [13]C-proline levels was observed between mock-AWF and flg22-AWF isolated from QKO leaves at any time point (Fig. 5c).

### PutA is required for *P. syringae* T3SS gene expression and virulence during infection of Arabidopsis

Proline utilization A (PutA) is a bacterial enzyme that catabolizes proline to glutamate (Fig. 6a)[22]. The expression of *putA* is induced by environmental proline[23]. To monitor proline perception by DC3000, we fused the promoter sequence of *putA* upstream of *gfp* and introduced this reporter construct into DC3000 to generate a DC3000 *putA_promoter:gfp* strain. We first tested the specificity of *putA* expression by culturing DC3000 *putA_promoter:gfp* in M9 medium supplemented with individual amino acids. Importantly, proline was the only amino acid tested that significantly induced *putA* expression (Fig. S4a). Using *putA* expression as a proxy for the levels of environmental proline, we cultured DC3000 *putA_promoter:gfp* in AWF samples and confirmed that flg22-AWF has decreased *putA*-inducing activity compared to mock-

AWF (Fig. S4b). To assess proline levels within the leaf apoplast, we syringe-infiltrated DC3000 *putA_promoter:gfp* into mock- and flg22-treated Arabidopsis leaves. Based on GFP fluorescence from the infiltrated leaf tissue, *putA* was rapidly induced in mock-treated leaves with significant induction detected 4.5 h after infiltration (Fig. S4c). In comparison, *putA* expression in flg22-treated leaves was significantly lower (Fig. S4d). Together, these results indicate that DC3000 encounters proline within the Arabidopsis leaf apoplast and that apoplastic proline levels are lower in flg22-treated tissue.

We next deleted *putA* from the DC3000 genome and tested the resulting Δ*putA* mutant strain for altered proline-induced responses and virulence. Similar to proline-specific expression of *putA*, DC3000Δ*putA* was unable to use proline as a growth substrate, but its ability to use other amino acids as growth substrates was unaffected (Fig. 6b, c). To assess the role of *putA* in T3SS induction, we introduced a *hrpL_promoter:gfp* reporter plasmid into DC3000Δ*putA* and cultured the resulting Δ*putA hrpL_promoter:gfp* strain in MM supplemented with proline or other known T3SS-inducing metabolites including aspartic acid and citric acid[9,10]. The induction of *hrpL* in response to proline, but

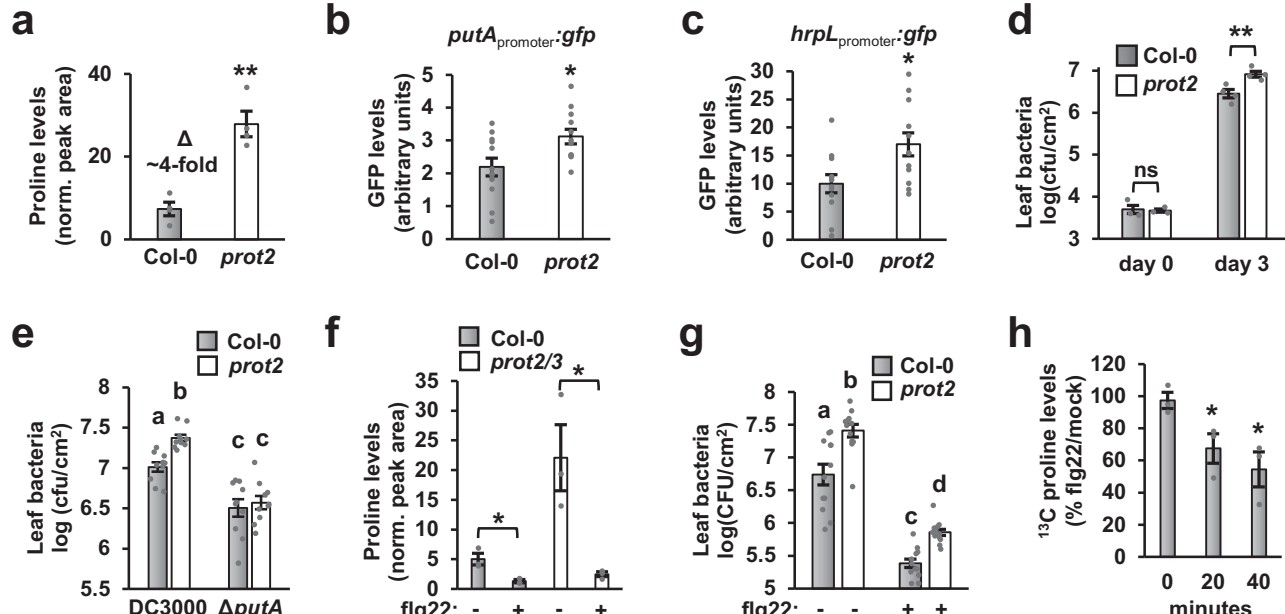

**Fig. 7 | Arabidopsis *prot2* leaves have elevated levels of apoplastic proline and are more susceptible to *P. syringae* infection. a** GC-MS analysis of proline in apoplastic wash fluid (AWF) from Col-0 and *prot2-3* leaves. Graphed are means ± SE of normalized peak area for proline, *n* = 4 from four independent experiments. Col-0 and *prot2-3* leaves were infiltrated with *P. syringae* DC3000 harboring either (**b**) *putA*promoter:*gfp* or (**c**) *hrpL*promoter:*gfp*, or an empty:*gfp* plasmid. Graphed are means ± SE of normalized GFP fluorescence of leaf tissue six hours post-infection, *n* = 12. **d** 2 x 10⁶ cfu/mL of *P. syringae* DC3000 was infiltrated into Col-0 and *prot2-3* leaves. Leaf bacteria were enumerated 3 days post-infection by serial dilution plating of leaf extracts. Graphed are log transformed means ± SE of colony-forming units (cfu), *n* = 3 for day 0 samples, *n* = 4 for day 3 samples. **e** 2 x 10⁶ cfu/mL of *P. syringae* DC3000 or DC3000Δ*putA* was infiltrated into Col-0 and *prot2-3* leaves. Leaf bacteria were enumerated 3 days post-infection by serial dilution plating of

leaf extracts. Graphed are log transformed means ± SE of colony-forming units (cfu), *n* = 12. **f** Proline levels in AWF from Col-0 and *prot2-3/prot3-2* leaves treated for six hours with 100 nM flg22 or a mock treatment. Graphed are means ± SE of normalized peak area for proline, *n* = 4. **g** *P. syringae* DC3000 growth in Col-0 or *prot2-3* plants pre-treated for six hours with 100 nM flg22 or a mock treatment. Graphed are log-transformed means ± SE of cfu isolated from infected tissue 3 days post-infection, *n* = 10. **h** ¹³C-proline uptake in *prot2-3* leaves treated for six hours with 100 nM flg22 or a mock treatment. Graphed are means ± SE of normalized peak area for proline, *n* = 3. Asterisks in panels **a–d**, **f** and **h** denote statistical significance based on two-sided *t*-test, **p* < 0.05. ***p* < 0.01. Lower case letters in panels **e** and **g** denote significance groupings based upon ANOVA with Tukey's HSD, *p* < 0.05. Data in panels **b–d** are representative of three independent experiments. Data in panels **e–h** were pooled from three independent experiments.

not other T3SS-inducing metabolites, was completely compromised in DC3000Δ*putA* (Fig. 6d). Next, we infiltrated DC3000 and DC3000Δ*putA* individually into Arabidopsis Col-0 leaf tissue, and detected significantly decreased AvrPto abundance and decreased bacterial growth in DC3000Δ*putA*-infected tissue (Fig. 6e, f). We also isolated a DC3000 *putA*::Tn5 strain and confirmed that the observed phenotypes are due to loss of *putA* (Fig. S4e–h). Taken together, these data show that proline is a T3SS-inducing signal within the Arabidopsis leaf apoplast and is essential for maximal DC3000 T3SS deployment and growth during infection.

### Arabidopsis *prot2* leaves have elevated levels of apoplastic proline and are more susceptible to *P. syringae* infection

We hypothesized that flg22-induced depletion of apoplastic proline may be due to altered activity of one or more plasma membrane-localized amino acid transporters. Among possible candidates, the transporters ProT1, ProT2 and ProT3 are known to transport proline, gamma-aminobutyric acid and glycine betaine[24]. We used GC-MS to measure proline levels in AWF isolated from leaves of Arabidopsis mutants lacking functional *ProT1*, *ProT2* or *ProT3* genes. Four-fold higher levels of proline were detected in AWF from *prot2-3* leaves but not in AWF from *prot1-1* or *prot3-2* leaves (Fig. 7a and Fig. S5a). To confirm that *prot2-3* leaves have increased levels of apoplastic proline, we infiltrated Col-0 and *prot2-3* leaves with DC3000 *putA*promoter:*gfp* and detected significantly higher levels of *putA* expression in the infected *prot2-3* tissue (Fig. 7b).

Next, we assessed the impact of increased apoplastic proline in *prot2-3* leaves on DC3000 T3SS deployment and virulence. In DC3000-

infected *prot2-3* leaves we detected significantly higher levels of *hrpL* expression, AvrPto accumulation and bacterial growth (Fig. 7c, d and Fig. S5b). Increased DC3000 growth also occurred in leaves of a mutant carrying an alternate non-functional *prot2-1* allele (Fig. S5d). In contrast, increased bacterial growth was not observed in DC3000-infected *prot1-1* or *prot3-2* (Fig. S5c). A similar increase in bacterial growth did not occur in *prot2-3* leaves infected with DC3000Δ*putA* (Fig. 7e), indicating that increased apoplastic proline is indeed causal for the enhanced virulence of DC3000 in *prot2* leaves. Together, these data reveal that ProT2 negatively regulates the levels of apoplastic proline in naïve Arabidopsis leaves, and demonstrate that apoplastic proline is a key virulence-inducing signal for DC30000 during infection.

In Arabidopsis, *ProT2* is rapidly induced by flg22 treatment[25], suggesting a possible role for ProT2 in flg22-induced depletion of apoplastic proline. However, flg22-dependent decrease in apoplastic proline and inhibition of DC3000 growth remained intact in *prot2-3/prot3-2* and *prot2-3* leaves, respectively (Fig. 7f, g). Furthermore, increased ¹³C proline uptake occurred in flg22-treated *prot2-3* leaves (Fig. 7h). Based on these data, we conclude that ProT2 is not required for flg22-induced depletion of apoplastic proline.

### The Arabidopsis amino acid transporter LHT1 is required for flg22-triggered depletion of apoplastic proline

We hypothesized that a transporter other than ProT2 may be responsible for the observed flg22-dependent depletion of apoplastic proline. We focused our efforts on LYSINE HISTIDINE TRANS-PORTER 1 (LHT1), a plasma membrane-localized protein that

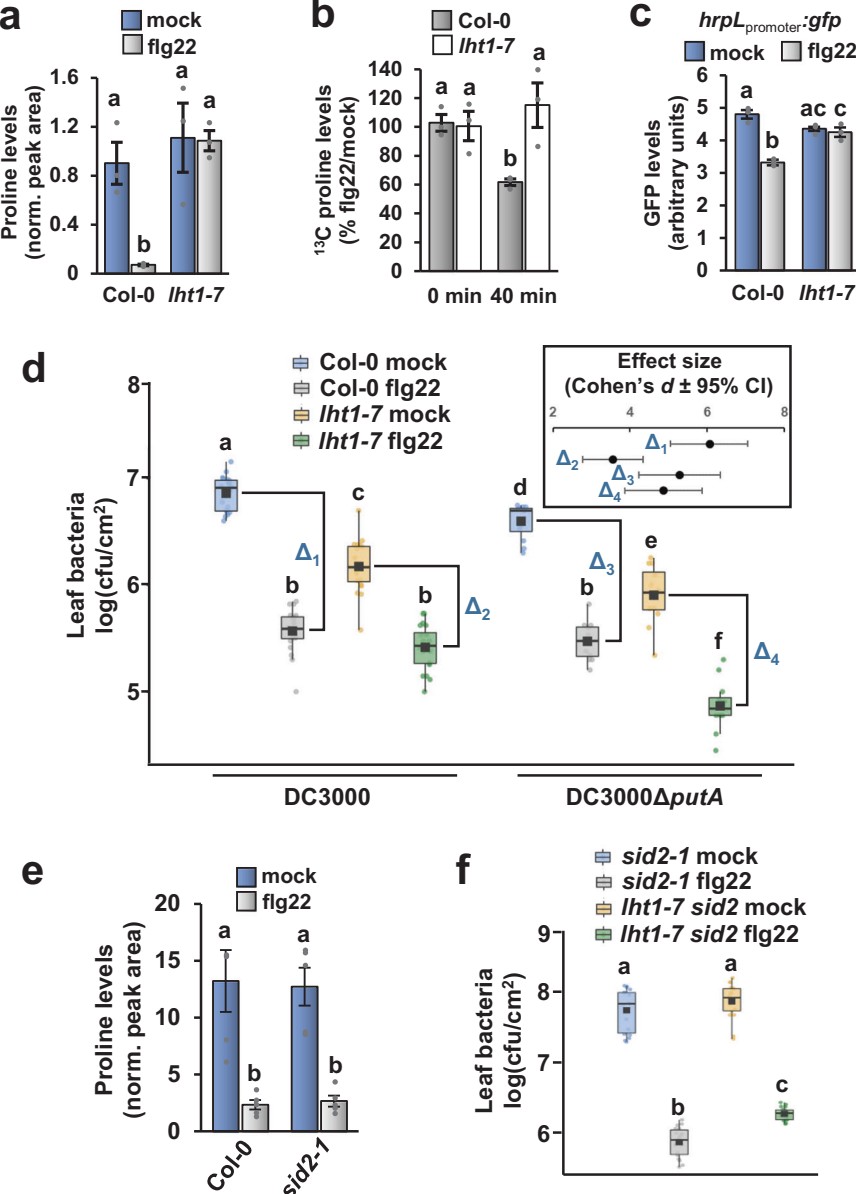

**Fig. 8 | Arabidopsis *LHT1* is required for flg22-induced depletion of apoplastic proline and maximal pattern-triggered immunity to *P. syringae*. a–c** Col-0 and *lht1-7* leaves were syringe-infiltrated with 100 nM flg22 or a mock treatment. **a** Proline levels in apoplastic wash fluid (AWF) isolated from leaves eight hours post-treatment. Graphed are means ± SE of normalized peak areas for proline, *n* = 3. **b** $^{13}$C-proline was infiltrated into treated leaves eight hours post-treatment, then AWF isolated from the treated leaves at timepoints indicated. Graphed are means ± SE of normalized peak area for $^{13}$C-proline in AWF, *n* = 3. **c** *P. syringae* DC3000 carrying a *hrpL*promotor*:gfp* or empty *gfp* plasmid were incubated for 10 h in AWF isolated from leaves eight hours post-treatment. Graphed are means ± SE of normalized GFP fluorescence, *n* = 3. **d** 1 × 10$^6$ cfu/mL of *P. syringae* DC3000 or DC3000Δ*putA* was syringe-infiltrated into Col-0 and *lht1-7* leaves pre-treated for six hours with 100 nM flg22 or a mock treatment. Leaf bacteria were enumerated on day 3 by serial dilution and plating of leaf extracts.

Graphed are log-transformed means ± SE of colony-forming units (cfu) of bacteria isolated from infected tissue, *n* = 12 and 20 for DC3000 Δ*putA* -and DC3000-infected samples, respectively. **e** Proline levels in apoplastic wash fluid (AWF) isolated from Col-0 and *lht1-7* plants eight hours after syringe-infiltration with 100 nM flg22 or a mock treatment. Graphed are means ± SE of normalized peak areas for proline, *n* = 5. **f** 1 × 10$^6$ cfu/mL of *P. syringae* DC3000 was syringe-infiltrated into *sid2-1* and *sid2-1/lht1-7* leaves pre-treated for six hours with 100 nM flg22 or a mock treatment. Leaf bacteria were enumerated on day 3 by serial dilution and plating of leaf extracts. Graphed are log-transformed means ± SE of colony-forming units (cfu) of bacteria isolated from infected tissue, *n* = 18. Lower case letters in all panels denote significance groupings based upon ANOVA with Tukey's HSD, *p* < 0.05. Box plots in **d** and **f** show median, interquartile range, min, max, black boxes are the mean values. Data in all panels were pooled from three independent experiments.

transports neutral and acidic amino acids including proline into Arabidopsis mesophyll cells[26–29]. *LHT1* is transcriptionally induced by biotic and abiotic stressors including flg22 treatment[25,29]. First, we measured proline levels in AWF isolated from leaves of a loss-of-function *lht1-7* mutant treated with flg22 or a mock solution. In contrast to the flg22-induced reduction in apoplastic proline in Col-0 leaves, we detected no significant difference in apoplastic proline

levels between mock- and flg22-treated *lht1-7* leaves (Fig. 8a). We observed the same phenotype with *lht1-5*, a mutant carrying an alternative loss-of-function allele (Fig. S6b). Using $^{13}$C-proline to directly assess the rate of proline depletion, we found that the flg22-induced depletion observed in Col-0 leaves did not occur in *lht1-7* leaves (Fig. 8b). To determine if the lack of apoplastic proline depletion in *lht1-7* impacts the expression of T3SS genes in DC3000,

we incubated DC3000 $hrpL_{promoter}$:*gfp* in AWF from mock or flg22-treated Col-0 and *lht1-7* leaves. No significant difference in *hrpL* expression was observed in DC3000 incubated in AWF from mock or flg22-treated *lht1-7* leaves (Fig. 8c). Together, these data show that LHT1 is required for flg22-induced depletion of apoplastic proline.

We hypothesized that LHT1-mediated depletion of apoplastic proline directly contributes to PTI against *P. syringae* infection. To test this hypothesis, we measured the growth of DC3000 and DC3000Δ*putA* in mock- and flg22-treated Col-0 and *lht1-7* leaves. Because DC3000Δ*putA* is specifically compromised in its ability to respond to proline (Fig. 6), the growth of this mutant relative to DC3000 provides a measure of apoplastic proline levels in infected leaf tissue and its impact on *P. syringae* virulence. Our hypothesis predicts three outcomes of these infection assays. First, because apoplastic proline is depleted in Col-0 in response to flg22, DC3000 and DC3000Δ*putA* should grow to similar levels in flg22-treated Col-0 leaves. Indeed, in flg22-treated Col-0 leaves, growth of DC3000 and DC3000Δ*putA* was indistinguishable (Fig. 8d, gray color). Second, because flg22 treatment does not alter levels of apoplastic proline in *lht1-7* leaves (Fig. 8a), sufficient levels of proline should remain in flg22-treated *lht1-7* leaves to function as a T3SS-inducing signal and promote DC3000 growth. Consistent with this prediction, in flg22-treated *lht1-7* leaves, DC3000 grew to significantly higher levels than DC3000Δ*putA* (Fig. 8d, green color). Third, the magnitude of flg22-induced suppression of DC3000 growth should be smaller in *lht1-7* plants if proline depletion from the apoplast contributes to PTI. In this regard, the effect size of flg22 treatment on DC3000 growth was significantly smaller in *lht1-7* leaves compared to Col-0 leaves (Fig. 8d, $\Delta_1$ vs. $\Delta_2$, inset). For DC3000Δ*putA*-infected plants, the presence or absence of *LHT1* did not significantly impact the effect size of flg22 treatment (Fig. 8d, $\Delta_3$ vs. $\Delta_4$, inset), indicating that the contribution of *LHT1* to PTI is linked to the presence of *putA* in DC3000.

Arabidopsis mutants lacking *LHT1* have constitutively increased salicylic acid-dependent resistance to *P. syringae*[30] and this enhanced resistance was observed in our experiments (Fig. 8d). To address whether SA is required for the contribution of *LHT1* to PTI against DC3000, we first measured apoplastic proline levels in mock- and flg22-treated leaves of Col-0 and an SA-deficient *sid2-1* mutant. No significant difference in flg22-induced depletion of apoplastic proline was observed in *sid2-1* compared to Col-0 (Fig. 8e). We then measured the growth of DC3000 in mock- and flg22-treated leaves of *sid2-1* and an *lht1-7 sid2-1* double mutant. DC3000 grew to similar levels in leaves of mock-treated *sid2-1* and *lht1-7 sid2-1* plants, confirming that constitutive resistance of *lht1* to *P. syringae* is SA-dependent (Fig. 8f). In contrast, in flg22-treated leaves, DC3000 grew to significantly higher levels in *lht1-7 sid2-1* compared to *sid2-1* (Fig. 8f), indicating that *LHT1* contributes to PTI in the absence of SA. These data are consistent with our observation that LHT1-dependent removal of proline is SA-independent (Fig. 8e). Based on these results, we conclude that LHT1-dependent removal of apoplastic proline directly contributes to flg22-induced PTI against *P. syringae* infection. Importantly, flg22 treatment significantly decreased the growth of DC3000 in *lht1-7 sid2-1* leaves (Fig. 8f). Therefore, additional *LHT1*- and SA-independent factors must contribute to flg22-induced resistance.

## Discussion

Pathogenic microbes deploy virulence factors to suppress host defenses and establish a habitable niche. In many cases, the production of virulence factors is induced by signals from the host environment[7]. Here, we investigated how flg22-induced changes in the Arabidopsis leaf apoplast restricts *P. syringae* from producing its T3SS. Through a combined metabolomics and genetics approach, we discovered that flg22-induced depletion of apoplastic proline, mediated by the amino acid transporter LHT1, directly contributes to restriction

of the *P. syringae* T3SS and PTI against *P. syringae* infection. These results uncover a mechanism of plant disease resistance, whereby depletion of a single extracellular metabolite, rather than accumulation of antimicrobial compounds or increased physical barriers, effectively limits the virulence of a bacterial pathogen.

Proline is involved in many cellular processes including regulation of osmotic pressure, redox potential and nutrient availability[31,32]. In plants, proline can be synthesized de novo or acquired from the soil, and can be transported through the vasculature, where its accumulation in specific tissues is intricately linked to developmental stage and nutrient status[33]. It is well established that proline accumulates in plant tissues in response to many biotic and abiotic stresses, primarily due to increased biosynthesis in stressed cells[34]. Yet, less is known about how levels of extracellular proline are regulated. Our data reveal that proline levels within the Arabidopsis leaf apoplast are highly dynamic and controlled by complex processes. In mock-treated leaf tissue, our $^{13}$C-proline probe was rapidly depleted within one hour from the apoplast of Col-0 leaves (Fig. 5a). Depletion of $^{13}$C-proline also occurred in mock-treated *prot2* and *lht1* leaves (Fig. S3c, d), suggesting ProT2 and LHT1 may act partially or redundantly to modulate apoplastic proline levels, or that additional membrane transporters may also contribute to proline uptake. In mock-treated tissue, loss of *PROT2* but not *LHT1* increased the total levels of apoplastic proline (Figs. 7a and 8a). These differences may be due to the relative affinities and/or transport rates of PROT2 and LHT1 for proline, or possibly altered activity of yet-to-be-identified efflux transporters responsible for loading proline into the apoplastic space.

In flg22-treated leaves we detected decreased levels of apoplastic proline as well as increased rates of $^{13}$C-proline depletion, and both of these phenotypes were completely dependent on *LHT1* (Fig. 8a, b). Furthermore, flg22-induced restriction of DC3000 growth was significantly compromised in *lht1* mutant leaves (Fig. 8d). These results suggest that MAMP perception may alter LHT1 abundance and/or transport activity, resulting in increased proline uptake. In this regard, *LHT1* transcripts accumulate within 30 min after flg22 treatment (Fig. S6a)[25], and this increase likely results in more LHT1 at the plasma membrane. However, in a previous study, constitutive over-expression of *LHT1* did not decrease proline levels within the Arabidopsis leaf apoplast[29]. Therefore, increased *LHT1* transcription may be insufficient to stimulate increased proline uptake. Alternatively, LHT1 activity may be post-translationally regulated, perhaps similar to phosphorylation-dependent regulation of NADPH oxidase RBOHD and sugar transporter STP13 by flg22-stimulated kinases[35,36].

Arabidopsis mutants lacking *LHT1* have increased salicylic acid-dependent resistance to *P. syringae* (Fig. 8d)[30]. Mutants lacking immune signaling components often have constitutive defense phenotypes[37–40], in some cases due to activation of intracellular immune receptors that "guard" immune signaling pathways, as exemplified by CSA1/CHS3 monitoring of the BAK1-BIR3 complex[41–43]. Because LHT1 contributes to PTI, it may be guarded in a similar manner. Heightened resistance of *lht1* may also be due to altered cellular redox status[30]. Regardless of the underlying mechanism(s) for constitutively elevated resistance, in *lht1* leaves we detected a significant decrease in the magnitude of flg22-induced inhibition of DC3000 growth (Fig. 8d). Furthermore, our experiments with SA-deficient *sid2* and *sid2 lht1* mutants revealed that *LHT1* significantly contributes to PTI against DC3000 infection even in the absence of SA (Fig. 8f). Together, these data demonstrate that *LHT1* functions in resistance against DC3000 in two genetically distinct ways. First, in the absence of any MAMP pre-treatment, *LHT1* negatively regulates SA-mediated defenses. Second, in MAMP-treated leaves, *LHT1* positively regulates a portion of PTI that is SA-independent. These conclusions, based on DC3000 growth data, are consistent with our observation that depletion of apoplastic proline in response to flg22 is SA-independent (Fig. 8e).

Flg22-induced depletion of apoplastic proline was completely compromised in the QKO mutant (Fig. 3e, f). Among the four genes that are mutated in QKO, *DDE2* and *EIN2* are required for JA- and ethylene-dependent responses, respectively, whereas *PAD4* is involved both SA-dependent and -independent defense signaling[44–48]. Because flg22-induced depletion of apoplastic proline was intact in *sid2* leaves, loss of *DDE2*, *EIN2* or *PAD4* individually may be causal for loss of apoplastic proline depletion in flg22-treated QKO leaves. Alternatively, PTI is robust to single mutant perturbation due to compensatory signaling through hormone-mediated defense pathways[17,49]. Therefore, SA, JA, ethylene, and PAD4 signaling sectors may coordinately regulate LHT1-dependent proline depletion. In future experiments, measuring apoplastic proline levels in flg22-treated leaves of mutants carrying higher order combinations of *sid2*, *dde2*, *ein2* and *pad4* mutant alleles may provide important clues about PRR-activated signaling pathways that regulate LHT1 activity.

The importance of host-derived proline for bacterial virulence has been previously demonstrated for both plant and animal pathogens[32,50,51]. For entomopathogenic *Photorhabdus*, high proline levels encountered within the insect hemolymph stimulates the production of virulence factors including insecticidal toxins[52], whereas for plant pathogenic *Agrobacterium*, host-derived proline in infected tissues promotes quorum sensing and the inter-bacterial exchange of virulence plasmids[50,51]. In DC3000, the requirement of *putA* for proline-induced *hrpL* expression suggests proline catabolism may be necessary for proline induction of the T3SS. In all organisms, proline is catabolized to glutamate which, in turn, can be converted to the TCA cycle intermediate α-ketoglutarate[53]. In many Gram-negative bacteria including pseudomonads, PutA catalyzes both of the enzymatic steps necessary to catabolize proline to glutamate[53]. Here we show that loss of *putA* in DC3000 completely abolishes proline-induced expression of *hrpL* and reduces DC3000 virulence (Fig. 6). Based on the predicted enzymatic functions of PutA, it is possible that one or both of the catabolites produced by PutA, rather than proline itself, may be T3SS-inducing signals. Alternatively, DC3000 PutA possesses a predicted DNA-binding domain that may directly or indirectly regulate the expression of T3SS-encoding genes. In support of this possibility, PutA in the plant pathogen *Ralstonia solanacearum* directly binds to the promoter of virulence-associated gene *epsA* and upregulates *epsA* expression in a proline-dependent manner[54].

In addition to proline, other metabolites such as aspartate and glutamate[9,10], as well as simple sugars such as fructose[8,55], also induce T3SS genes in DC3000 and were detected in our AWF samples (Fig. 3). Similar to phenotypes of DC3000 Δ*putA*, DC3000 mutants that are less responsive to these host metabolite signals are also less virulent[8,9]. Together, these observations suggest that, in naïve leaf tissue, DC3000 encounters a mixture of T3SS-inducing metabolites, with each metabolite quantitatively contributing to observed levels of T3SS induction and virulence (Fig. S7). In support of this model, a proline-insensitive DC3000 Δ*putA* strain is more virulent than a T3SS-deficient DC3000 *hrcC⁻* strain (Fig. S4i), indicating that additional proline-independent signals must be present in the apoplast. In our metabolomics analysis, the abundance of T3SS-inducing aspartate, glutamate and simple sugars did not change in the apoplast in response to flg22 treatment (Fig. 3). These metabolites are likely responsible for residual T3SS inducing activity observed in flg22-AWF (Fig. 4c). Despite the presence of these additional T3SS-inducing signals, we propose that depletion of proline from the apoplast is sufficient to decrease the magnitude of T3SS induction below a threshold level required for maximal virulence (Fig. S7).

Previous studies have investigated the role of extracellular amino acids and sugars in PTI. Flg22-treated Arabidopsis seedlings had enhanced uptake of glucose and fructose[36], and exuded lower amounts of sixteen amino acids[56]. These changes in sugars and amino acids required *STP13* and *LHT1*, respectively[36,56]. Additionally,

Arabidopsis *mkp1* seedlings and MAMP-treated Arabidopsis suspension cell cultures exuded lower levels of many metabolites including several T3SS-inducing organic acids and amino acids[10,57]. In this study we did not observe similar large-scale changes in amino acid and sugar levels within the apoplast of flg22-treated leaves, suggesting that MAMP-induced changes to the exometabolome of seedlings and suspension cells are distinct from those that occur within the leaf apoplast. Though, a caveat to this conclusion is that the single time point in our metabolomics analysis likely does not capture the full spectrum of MAMP-induced changes that occur. In this regard, a recent study reported that nine different amino acids significantly accumulated within the apoplast of Arabidopsis leaves 24 h after flg22 treatment[58]. Because high concentrations of amino acids can repress T3SS genes in *P. syringae*, the authors concluded that increased amino acid levels within the apoplast contributes to flg22-induced T3SS repression and PTI[58]. However, these late-occurring changes are unlikely to be causal for T3SS repressive environment that is established within ten hours after MAMP perception (Fig. 1)[12]. Additionally, in light of recent evidence that some type III effectors promote the extracellular release of nutritive metabolites[59,60], it is not clear why increasing the amino acid levels within the apoplast would be an effective mechanism of resistance. A detailed time course analysis of MAMP-induced changes to the metabolic composition of the apoplast, including during *P. syringae* infection, will be necessary to understand the full scope of metabolic changes that occur.

The primary objective of this study was to understand how PTI restricts T3SS deployment. From this perspective, our data fully support a model wherein depletion of apoplastic proline as a T3SS-inducing signal directly contributes to PTI against *P. syringae*. However, our data also reveal that proline is likely a nutrient for *P. syringae* during infection (Fig. 4d), and it is clear from our experiments with DC3000 Δ*putA* that proline catabolism and T3SS gene induction are intricately linked (Fig. 6). Because of these intertwined functions, it is not possible to discern from our data whether loss of proline as a nutrient source also contributes to decreased growth of DC3000 in MAMP-treated tissues. For many pathogens, successful infection requires a period of rapid pathogen growth that is fueled by host-produced nutrients[61,62]. Thus, it is logical that restricting access to nutrients in the apoplast could be an effective means to limit infection by many pathogens, as previously proposed[10,26,29,61,63]. In addition to PTI, effector-triggered immunity (ETI) also restricts *P. syringae* from delivering type III effectors[64], and a recent study reported that *LHT1* is required for ETI initiated by RPS2 recognition of effector AvrRpt2[65]. Whether LHT1-mediated changes to the apoplast metabolome also contribute to ETI will be important to address in future work. From a broader perspective, nutrient exchange is fundamental to plant-microbe symbioses, including those with non-pathogenic endophytes[66]. Furthermore, plant innate immunity and the establishment of communities of plant microbiota appear to be intimately linked[67,68]. In this light, whether PRR-mediated restriction of apoplastic proline impacts plant-associated commensal and beneficial microbes, as well as diseases caused by other pathogenic microbes, is an interesting question for future research.

## Methods

### Preparation of peptide stocks
Flg22 and elf26 peptides were synthesized by Genscript and stored at −20 °C as 1 mM stocks in DMSO.

### Plant material and growth conditions
Arabidopsis seeds were sterilized, stratified and sown on the surface of MS agar as described previously[10]. The seeds were placed in a Percival CU41L4 growth chamber with environmental conditions set to 40% light strength, 22.5 °C/21 °C day/night temperatures, and a 10 h day cycle. After 10 days, seedlings were transplanted to Sunshine Mix soil

(Sun Gro Horticulture) and grown for 3 to 4 weeks in the same Percival chamber under the same environmental conditions. The *fls2* (SALK_093905)[69], *efr-2* (SALK_068675)[16], *dde2 ein2 pad4 sid2*[17], *lht1-5* (SALK_115555C)[30], *lht1-7* (SALK_083700C)[30] and *sid2-1*[46] alleles are in the Col-0 ecotype background. The *lht1-7 sid2-1* mutant was generated by crossing of *lht1-7* and *sid2-1*. The homozygous *lht1-7 sid2-1* mutant was identified in the F2 generation by PCR genotyping for wild-type and mutant alleles. The *prot1* (SALK_018050), *prot2-3* (SALK_067508), and *prot3-2* (SALK_083340) mutants are in the Col-0 background, and the *prot2-2* (CSJ1230) mutant is in the Wassilewskija ecotype background[70].

### Construction of GFP transcriptional reporter plasmids

Construction of the reporter plasmid *hrpL*<sub>promoter</sub>:*gfp*::pProbe-NT was described previously[57]. To construct the *putA*<sub>promoter</sub>:*gfp*::pProbe-NT reporter plasmid. the oligonucleotides 5′-actctagaggatccccGTT GTTGGTGGAGCGATAC-3′ and 5′- ttcgagctcggtacccGTATTGTCCTC ATTGTAGCCAC-3′ were used to PCR amplify a DNA fragment corresponding to 643 bp upstream of the DC3000 *putA* coding region, and the Gibson assembly method used to clone the amplified PCR product into SmaI-digested pProbe-NT.

### Generation of DC3000 Δ*putA* and *putA*::Tn*5* mutants

The DC3000 *putA* deletion mutant was generated by suicide vector-mediated allelic exchange. Oligonucleotides 5′-ttcgagctcggtacccCAG GCCGGACATGTAGATAG-3′, 5′-atcatcctatcgtcatAGTGGTGGTGGCC ATGTATTG-3′, 5′-catggccaccaccactATGACGATAGGATGATGCGATAG-3′ and 5′-actctagaggatccccGCTGGCAGATGACAAATACG-3′ were used to PCR amplify two ~750 bp DNA fragments corresponding to regions immediately upstream and downstream of the *putA* coding sequence. The Gibson method was used to assemble the two fragments into SmaI-digested pK18*mobsacB*. The assembled plasmid was introduced into DC3000 by triparental mating and a double recombinant Δ*putA* strain isolated as described previously[9]. The *putA*::Tn*5* mutant strain was identified from a collection of DC3000 strains that were randomly mutagenized by transposon Tn5 as described previously[8]. The Tn*5* insertion site was confirmed by PCR-based genotyping.

### Preparation of *P. syringae* inoculum

*P. syringae* pv *tomato* DC3000 strains were maintained at −80 °C as frozen suspensions in 20% glycerol. Prior to use, the bacteria were streaked onto a modified King's B (KB) medium[8] solidified with 1.5% agar and supplemented with the appropriate antibiotics for both strain and plasmid selection. For experiments that included Tn*5*::*putA* or Δ*putA* strains, the bacteria were grown on KB agar for two days at room temperature, then inoculated into 40 mL of KB broth supplemented with the appropriate antibiotics. The starting optical density at λ = 600 nm (OD<sub>600</sub>) of the cultures was 0.1. The cultures were placed into a 250 rpm shaking incubator at 28 °C for 7 h, then centrifuged at 10,000 x *g* for 10 min. After discarding the supernatant, the pelleted bacteria were washed in one mL of sterile H₂O prior to use. For experiments that did not include DC3000 Tn*5*::*putA* or Δ*putA* strains, the bacteria were grown on KB agar for two days at room temperature, scraped off the surface of the agar plate, resuspended in one mL of sterile water, and washed three times with sterile water prior to use.

### GFP transcriptional reporter assays

Measurements of GFP fluorescence from liquid cultures of DC3000 carrying *hrpL*<sub>promoter</sub>:*gfp*::pProbe-NT or *putA*<sub>promoter</sub>:*gfp*::pProbe-NT reporter plasmids was done as described previously[9]. For measurements of *hrpL* and *putA* expression in DC3000-infected leaf tissue, a needle-less syringe was used to infiltrate fully expanded leaves of 4- to 5-week-old plants with sterile H₂O containing either 100 nM flg22 or a DMSO control. After five hours, the same leaves were syringe-

infiltrated with an OD<sub>600</sub> = 0.5 inoculum of DC3000 carrying *hrpL*<sub>promoter</sub>:*gfp*::pProbe-NT, *putA*<sub>promoter</sub>:*gfp*::pProbe-NT, or an empty vector pProbe-NT plasmid. After three hours, twelve 0.3 cm² leaf disks were excised from tissue infected with each strain and GFP fluorescence from each leaf disk measured as described previously[8].

### Detection of AvrPto in *P. syringae*-infected leaf tissue

Fully expanded leaves of 4- to 5-week-old plants were syringe-infiltrated with sterile water containing either 100 nM flg22, 100 nM elf26 or an equivalent amount of DMSO as a solvent only control. Five hours after infiltration, the same leaves were syringe-infiltrated with an OD<sub>600</sub> = 0.5 solution of DC3000. After five hours, four 0.3 cm² leaf disks were collected from infected tissue and frozen at −80 °C. Immunoblot detection of AvrPto in the collected leaf tissue was done as described previously[10].

### Measurement of DC3000 growth in Arabidopsis leaves

For measurement of DC3000 growth *in planta*, a needle-less syringe was used to infiltrate an OD<sub>600</sub> = 0.001 solution of DC3000 into fully expanded leaves of a 4- to 5-week-old Arabidopsis plants. DC3000 levels in the infected tissue were measured by serial dilution plating of leaf extracts as described previously[71].

### Isolation of apoplastic wash fluid from Arabidopsis leaves

Apoplastic wash fluid (AWF) was isolated from leaves of 5- to 6-week-old Arabidopsis plants treated with flg22 or a mock control. To initiate MAMP responses, a needle-less syringe was used to infiltrate leaves with a solution of 100 nM flg22 in water, or with DMSO in water as a negative control. Six to eight leaves were infiltrated on each plant, and a total of six plants were infiltrated for each treatment. After eight hours, AWF was isolated by syringe-infiltrating the mock- and flg22-treated leaves with sterile H₂O containing 164 μM ribitol. Immediately after infiltration, the aerial portion of the plant was removed by cutting the primary stem and briefly washed with H₂O to remove surface contaminants. The infiltrated leaves were detached from the rosette and stacked between layers of parafilm. The parafilm booklet of leaves was wrapped with tape and suspended inside a 15 mL conical centrifuge tube. The tube was centrifuged at 750 x *g* for seven minutes. The AWF that collected at the bottom of the tube was transferred to a microcentrifuge tube, then centrifuged at 21,000 x *g* for 10 minutes at 4 °C. The resulting supernatant was transferred to a microcentrifuge tube. After addition of 50 μL of chloroform, the samples were vortexed for 10 seconds and centrifuged at 21,000 x *g* for 10 min at 4 °C. The upper aqueous phase was transferred to a microcentrifuge tube, and the volume recovered was measured with a pipette. The AWF samples were then lyophilized to dryness and stored at −80 °C. Malate dehydrogenase (MDH) activity in AWF was measured as described previously[72]. To separate the contents of AWF into water- and chloroform-soluble fractions, 50 μL of chloroform was added to 300 μL of isolated AWF. The mixture was briefly vortexed and centrifuged at 21,000 x *g*. The resulting aqueous and organic phases were transferred to separate tubes, frozen at −80 °C, lyophilized to dryness, and resuspended in 300 μL of sterile H₂O.

### Measurements of *P. syringae* T3SS deployment and growth in apoplastic wash fluid

Lyophilized apoplastic wash fluid (AWF) was resuspended in a minimal medium (MM) [10 mM K₂HPO₄/KH₂PO₄ (pH 6.0), 7.5 mM (NH₄)₂SO₄, 3.3 mM MgCl₂, 1.7 mM NaCl] to the original volume of AWF prior to lyophilization. For measurements of AvrPto abundance, 180 μL of the resuspended AWF was combined with 20 μL of an OD<sub>600</sub> = 1.0 solution of DC3000 within a single well of a 24 well polystyrene assay plate. The assay plate was incubated at room temperature on a shaking platform rotating at 130 rpm. After 6 h, the bacteria were transferred into a microcentrifuge tube and pelleted by

centrifugation at 21,000 x *g* for 10 min. The supernatant was removed and the bacterial pellet was frozen in liquid nitrogen. Immunoblot detection of AvrPto in the pelleted bacteria was done as described previously[10]. For measurements of *hrpL* and *putA* promoter activity and DC3000 growth in AWF, 40 µL of the resuspended AWF and 10 µL of an $OD_{600}$ = 0.5 solution of DC3000 carrying either *hrpL*promoter:*gfp*::pProbe-NT, *putA*promoter:*gfp*::pProbe-NT or empty:*gfp*::pProbe-NT were mixed within a single well of a 384 well assay plate. A Tecan Spark 10 M plate reader was used to measure GFP fluorescence and $OD_{600}$ of the cultures in each well as described previously[9]. GFP fluorescence measurements were first normalized to $Abs_{600}$ and then to fluorescence from wells containing a DC3000 empty:*gfp*::pProbe-NT strain under the same treatment conditions.

### Measurement of *P. syringae* growth in defined media
Growth of *P. syringae* in defined liquid media was measured in a 96-well plate format. A volume of 90 µL of KB broth, a modified M9 minimal medium[9] lacking glucose and ammonium chloride, or the same modified M9 minimal medium supplemented with 10 mM of an individual amino acid was mixed with 10 µL of an $OD_{600}$ = 1.0 inoculum of DC3000, DC3000 Δ*putA* or DC3000 *putA*::Tn5. A Tecan Spark 10 M plate reader was used to take $OD_{600}$ measurements of cultures in each well every 30 min. Between readings, the plate was shaken at 216 rpm and maintained at ambient temperature within the plate reader. A humidity cassette was used to limit evaporation from plate wells during the timecourse.

### GC-MS detection of metabolites in apoplastic wash fluid
For each sample, a 20 µL aliquot of the aqueous phase from chloroform-extracted AWF was lyophilized to dryness and resuspended in 10 µL of 30 mg/mL methoxyamine hydrochloride (Sigma-Aldrich) in pyridine (Sigma-Aldrich). The resuspended sample was incubated at 37 °C and shaken at 1800 rpm for 90 minutes. After adding 20 µL of N-methyl-N-(trimethylsilyl)trifluoroacetamide (MSTFA) with 1% trimethylchlorosilane (CovaChem), the sample was incubated at 37 °C and shaken at 1800 rpm for 30 minutes. Each derivatized sample was injected with a 10:1 split into an Agilent 7890B GC system with a 30 m + 10 m Duraguard x 0.25 mm x 0.25 µm DB-5MS + DG Agilent column. The oven temperature was kept at 60 °C for 1 min, then ramped to 300 °C at a rate of 10 °C/min and held at 300 °C for 10 min. Analytes were detected with an Agilent 5977B MSD in EI mode scanning from 50 m/z to 600 m/z. Mass spectrum analysis, component identification and peak area quantification were performed with AMDIS. Statistics were performed with MetaboAnalyst[73] and MetaboAnalystR[74]. Two to three technical replicates were analyzed for each sample and averaged to produce a single sample value. Absolute concentrations of metabolites in AWF were determined by spiking known concentrations of purified compounds into AWF samples and analyzing the resulting change in GC-MS peak areas. Peak areas for each external standard were normalized by the internal standard ribitol, and a standard curve was generated and fit with a linear trend line. The normalized peak areas obtained for the metabolites in AWF samples were converted into concentrations by using the slope of the line calculated from the standard curve and the average normalized peak area abundances measured in the AWF metabolomics analysis±SE.

### Measurements of ¹³C-proline uptake in leaves
Fully expanded Arabidopsis leaves were syringe-infiltrated with either 100 nM flg22 or a DMSO only control. After six to eight hours, a solution of 164 µM ribitol and 500 µM ¹³C L-proline was syringe-infiltrated into the same leaves and the plants were placed into a humidity chamber to prevent evaporation from the infiltrated leaves. AWF was extracted from the infiltrated leaves as described above either immediately or after 20, 40 or 60 min. The peak area of ¹³C

L-proline was measured with AMDIS using a custom library entry of the fragmentation pattern and retention time of a ¹³C L-proline standard.

### Quantitative real-time PCR (qRT-PCR) transcript analysis
Leaves of 4- to 5-week-old Col-0 plants were syringe-infiltrated with sterile water containing either 100 nM flg22 or a DMSO only control. After 40 min, TRIzol (Thermo Fisher Scientific) was used to isolate total RNA from the infiltrated leaf tissue. The isolated RNA was reverse transcribed and qRT-PCR was performed as previously described[11]. The abundance of *AT2G28390* transcripts measured in each sample was used for normalization. The following sequences were used as oligonucleotides for qRT-PCR to measure transcripts from the indicated genes: *AT2G28390*, 5′-AACTCTATGCAGCATTTGATCCACT-3′ and 5′-TGATTGCATATCTTTATCGCCATC-3′; *AT5G40780* (*LHT1*), 5′-CGT TGAAATCGGTGTTTGCATCGT-3′ and 5′-GCGATTGTTGAGTAGCTGA-GAGAC-3′.

### Statistics
Statistical analyses of data were done using Microsoft Excel, Metaboanalyst 5.0[73], R with the MetaboAnalyst for R package[74], and Jamovi software[75]. For all experiments *n* equals the number of measurements from distinct samples. All *t*-tests were two-tailed with no correction for multiple comparisons.

### Reporting summary
Further information on research design is available in the Nature Portfolio Reporting Summary linked to this article.

### Data availability
Raw GC-MS data files are deposited in the National Metabolomics Data Repository (www.metabolomicsworkbench.org) as study number ST002917. All other source data are provided in Supplementary Information/Source Data file. Source data are provided with this paper.

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

## Acknowledgements
We thank Scott Peck and Jeff Chang for helpful discussions and critical reading of this manuscript. This work was supported by National Science Foundation CAREER grant IOS-1942898 awarded to J.C.A. and Swiss National Science Foundation grants SNSF 3100-064918 and 31003A_149229 awarded to D.R.

## Author contributions
J.A. and C.R. conceived the project; C.R., Y.Y.P., S.M., S.T. and A.W. performed the experiments; J.A., C.R., Y.Y.P. and A.W. analyzed the data, D.R. and S.L. generated and provided mutant lines and shared unpublished data; J.A. and C.R. wrote an initial draft; J.A., C.R., Y.Y.P, S.M., S.T., S.L. and D.R. reviewed and edited the manuscript.

## Competing interests
The authors declare no competing interests.
