## [Peer Review File · Nature Communications]

REVIEWER COMMENTS

Reviewer #1 (Remarks to the Author):

Rogan et al. demonstrate the importance of apoplastic Pro for virulence induction in *P. syringae* during infection of *Arabidopsis* leaves. They also demonstrate the role of the plant uptake transporter LHT1 in mediating the PAMP (MAMP)-induced depletion of Pro from the apoplast. The findings advance our understanding of the relationship between host defense, apoplast composition, and bacterial virulence induction in planta. The conclusions are supported by a series of well controlled and executed experiments and clearly described. I offer a few suggestions below.

Major comments:

1. The authors interpret the data in figure 7D to conclude that the role of proline uptake by LHT1 contributes to resistance distinctly from the previously described activation of SA-mediated resistance in the *lht1* mutant. However, it is complicated to compare effect sizes between log-transformed data for different plant:bacteria combinations when the absolute bacterial numbers vary across an ~100-fold range. Double mutant plants with both *lht1* and second mutations that disrupt SA-signaling are available. The authors should determine if *flg22* induces Pro uptake in these SA-signaling mutant backgrounds (whether this will occur is not clear from the results with the quadruple QKO mutant that disrupts SA-, JA-, and Et-signaling). If not, they will have linked SA-signaling directly to the PTI-induced Pro uptake, which would be a nice addition to the findings. If so, including *lht* and SA-deficient double mutant plants in an experiment like figure 7D will enable a less complicated challenge to the model.
2. The robust Pro-dependent shifts in virulence expression and bacterial growth are surprising given that multiple elicitors of virulence expression are known to be present in the apoplast. It is hypothesized (figure S7) that Pro augments virulence expression above the level induced by these other elicitors. The authors are positioned (perhaps pending quantification of a few additional metabolites in the AWF) to challenge this model by examining induction of *hrpL* and/or accumulation of *AvrPto* in vitro in the presence of a cocktail of the known inducers +/- Pro. If the Pro-dependent expression is apparent in the cocktail, this would better support the model and thus strengthen the overall conclusions of the paper.

Minor comments:

3. It is suggested to use MAMP (microbe-associated...), rather than PAMP, since pathogenic and non-pathogenic microbes alike carry elicitors of pattern-recognition receptors.
4. Repetitive content in lines 11 to 14.
5. The authors should be more cautious about directly linking phenotypes in the QKO quadruple mutant to PTI-deficiency. The QKO plants may have additional deficiencies in non-PTI aspects of basal defense as well as in effector triggered immune responses, such as induced by CAR1 in response to *AvrE1* carried by DC3000.
6. Line 180: this experiment does not rule out uptake into the leaf vasculature where it then "protected" from centrifugation-based isolation of AWF. Perhaps note this caveat.

7. Line 200: the experiment in figure S4b indicates that the AWF from flg22-treated plants has decreased putA-inducing activity. Proline levels were not measure here.

8. Line 284-285: What is the distinction between “reduction in bacterial growth” and “the absolute level of resistance”? One could argue that bacterial growth is the best indicator of the level of resistance. Consider this section.

Reviewer #2 (Remarks to the Author):

The study by Rogan et al presents new findings addressing the regulation of apoplastic proline levels with respect to PTI against *Pseudomonas syringae*. Plant-exuded proline serves as nutrient for *Pseudomonas* and the respective uptake from the apoplast via the plant transporters ProT2 and LHT1 might serve as tool contributing to plant induced immunity.

The study is very well designed, with a properly controlled setup of experiments. The results are trustworthy. Besides, the manuscript is written in a very nice, coherent style. The results are of significance for the field of plant defense and connect to previous studies where amino acid transport was coupled to plant defense responses.

The only comment that I have deals about Figure 7 and the fairly quick turn to the amino acid transporter LHT1. Some more experiments would be needed to justify the drawn conclusions about the central involvement about the transporter:

1. Have the authors also checked the involvement of other transporters such as some of the AAPs, that can also contribute to proline transport? (line 334)
2. Also, it would be interesting to see the response in a *lht1*prot2* double mutant line. (line 335)
3. The authors could support their hypothesis by looking at published phospho-proteomic data and check if LHT1 has been found being phosphorylated. (line 352)
4. Did the authors check for regulation of Prot2 / LHT1 with respect to flg22-treatment?
5. The authors cite that overexpression of LHT1 does not lead to decreased proline levels in the apoplast. It would be nice to include this line in the experimental setup as control. How is it for Prot2?

Minor comment:

6. There are a lot of partially unnecessary abbreviations, e.g. line 105: MM and those abbreviations are

not consequently followed, e.g. line 196, minimal media.

Besides, I have no further comments and I thank the authors for a very interesting study!

Reviewer #3 (Remarks to the Author):

The authors study the role of flg2 on PTI induction, for that they analyse the extracellular fluids of Arabidopsis leaves after elicitation with the flg2. In a metabolomics approach the authors have evidences that point out for the specific accumulation or depletion of several small compounds and test some for the hrpL-inducing activity. They focus then on proline and conduct several experiments to prove that proline presents a relevant role in the infection outcome. The authors also look into some proline transporters to access their role in the proline depletion seen and suggest that LHT1 may be a player on this regulation. Overall the approach is very interesting and important however in some cases it is difficult to follow the authors rational.

I also believe that some of the conclusions that are drawn are not supported by the approach/results (see comments in the MS). The authors should assess these questions before the manuscript is considered for publication at Nature Communications.

REVIEWER COMMENTS

Reviewer #1 (Remarks to the Author):

Rogan et al. demonstrate the importance of apoplastic Pro for virulence induction in *P. syringae* during infection of Arabidopsis leaves. They also demonstrate the role of the plant uptake transporter LHT1 in mediating the PAMP (MAMP)-induced depletion of Pro from the apoplast. The findings advance our understanding of the relationship between host defense, apoplast composition, and bacterial virulence induction in planta. The conclusions are supported by a series of well controlled and executed experiments and clearly described. I offer a few suggestions below.

Major comments:

1. The authors interpret the data in figure 7D to conclude that the role of proline uptake by LHT1 contributes to resistance distinctly from the previously described activation of SA-mediated resistance in the *lht1* mutant. However, it is complicated to compare effect sizes between log-transformed data for different plant:bacteria combinations when the absolute bacterial numbers vary across an ~100-fold range. Double mutant plants with both *lht1* and second mutations that disrupt SA-signaling are available. The authors should determine if *flg22* induces Pro uptake in these SA-signaling mutant backgrounds (whether this will occur is not clear from the results with the quadruple QKO mutant that disrupts SA-, JA-, and Et-signaling).

Thank you for this suggestion. We measured Pro levels in apoplastic wash fluid (AWF) isolated from mock- and *flg22*-treated wild type Col-0 and SA-deficient *sid2* leaves. These data are now included as Figure 7e. No significant difference was detected in the magnitude of *flg22*-induced Pro reduction in AWF from *sid2* leaves compared to AWF from Col-0 leaves. These data demonstrate that *flg22*-induced removal of Pro from the apoplast is SA-independent, and are consistent with our new pathogen growth data described below that show the contribution of *LHT1* to PTI is independent of SA (Figure 7f).

If not, they will have linked SA-signaling directly to the PTI-induced Pro uptake, which would be a nice addition to the findings. If so, including *lht* and SA-deficient double mutant plants in an experiment like figure 7D will enable a less complicated challenge to the model.

We generated an *lht1/sid2* double mutant and tested this mutant, along with *sid2* plants, for *flg22*-induced immunity to DC3000. These data are included as Figure 7f. In these experiments, DC3000 grew to similar levels in mock-treated *sid2* and *lht1/sid2* plants, confirming that elevated resistance of *lht1* to *P. syringae* is SA-dependent. In contrast, in *flg22*-treated leaves, DC3000 grew to significantly higher levels in *lht1/sid2* compared to *sid2*. These data are consistent with our observation that *LHT1*-dependent removal of proline is SA-independent (Figure 7e) and are consistent with Col-0 and *lht1* infection assays showing *LHT1* contributes to PTI against DC3000 (Figure 7d).

2. The robust Pro-dependent shifts in virulence expression and bacterial growth are surprising given that multiple elicitors of virulence expression are known to be present in the apoplast. It is hypothesized (figure S7) that Pro augments virulence expression above the level induced by these other elicitors. The authors are positioned (perhaps pending quantification of a few

additional metabolites in the AWF) to challenge this model by examining induction of hrpL and/or accumulation of AvrPto in vitro in the presence of a cocktail of the know inducers +/- Pro. If the Pro-dependent expression is apparent in the cocktail, this would better support the model and thus strengthen the overall conclusions of the paper.

Thank you for this suggestion. We know from our GC-MS analyses that the T3SS-inducing compounds Asp, Glu and citric acid are present in AWF. We measured the absolute concentration of these metabolites in AWF. In mock-AWF, [Glu] was $432 \pm 102 \mu\text{M}$, [Asp] was $59 \pm 7 \mu\text{M}$ and [citric acid] was $304 \pm 32 \mu\text{M}$; whereas in flg22-AWF [Glu] was $459 \pm 103 \mu\text{M}$, [Asp] was $86 \pm 20 \mu\text{M}$ and [citric acid] was $292 \pm 56 \mu\text{M}$. Based on our previous studies, these concentrations of metabolites within AWF are within the range of concentrations necessary for T3SS-inducing bioactivity [Anderson et al. (2014) *PNAS* 111, 6846; Yan et al. (2020) *PLOS Pathog.* 16, e1008680]. Therefore, by adding proline back to AWF (Fig 3i), we have in effect done the experiment proposed here. We have not yet attempted to build a synthetic medium that mimics AWF because we do not know the full spectrum of T3SS-inducing metabolites that are present in AWF. Furthermore, there may be T3SS-repressing signals present that contribute to the total observed bioactivity of AWF. Because of these uncertainties, our opinion is that adding proline back to AWF (as done in Fig 3i) is currently the best experiment to assess if decrease proline is causal for decreased *P. syringae* T3SS and growth in flg22-AWF. We note that our experiments with DC3000 *putA* (Fig 5 and Fig S4) and *prot2* plants (Fig 6 and Fig S5) provide strong support for our conclusion that, despite the presence of other T3SS-inducing signals, proline is a virulence signal within the Arabidopsis apoplast. In past work we also identified DC3000 mutants that have decreased responses to T3SS-inducing Asp/Glu and sugars, and these mutants were similarly less virulent on Arabidopsis [Anderson et al. (2014) *PNAS* 111, 6846; Yan et al. (2020) *PLOS Pathog.* 16, e1008680]. To test the model in Fig S7, a longer term goal is to generate a DC3000 polymutant strain that is compromised in perception of multiple host signals, as a means to assess whether host signals quantitatively contribute to virulence as proposed. Though, these experiments will take substantial time and effort and are outside of the scope of this manuscript.

Minor comments:

3. It is suggested to use MAMP (microbe-associated...), rather than PAMP, since pathogenic and non-pathogenic microbes alike carry elicitors of pattern-recognition receptors.

Thank you, we revised to MAMP throughout manuscript. Because the acronym PAMP is commonplace in the literature, particularly in animal immunity literature, we have included “PAMP” as an alternate acronym within the introduction (line 5).

4. Repetitive content in lines 11 to 14.

Revised to remove repetitive content.

5. The authors should be more cautious about directly linking phenotypes in the QKO quadruple mutant to PTI-deficiency. The QKO plants may have additional deficiencies in non-PTI aspects of basal defense as well as in effector triggered immune responses, such as induced by CAR1 in response to AvrE1 carried by DC3000.

Thank you for pointing this out. We modified text in the paragraph starting at line 70 to define QKO as a mutant that is generally deficient in plant defense responses.

6. Line 180: this experiment does not rule out uptake into the leaf vasculature where it then "protected" from centrifugation-based isolation of AWF. Perhaps note this caveat.

Thank you for this suggestion. We revised this sentence, at line 180 in revised manuscript, to: “A similar rate of ¹³C-proline depletion was observed in AWF from leaves detached from the rosette, thus ruling out long distance vascular transport contributing to ¹³C-proline depletion, though these data do not rule out local uptake of ¹³C-proline into vascular tissues (Fig. S3b).”

7. Line 200: the experiment in figure S4b indicates that the AWF from flg22-treated plants has decreased putA-inducing activity. Proline levels were not measure here.

Revised to “confirmed that flg22-AWF has decreased putA-inducing activity compared to mock-AWF.” (line 202 in revised manuscript).

8. Line 284-285: What is the distinction between “reduction in bacterial growth” and “the absolute level of resistance”? One could argue that bacterial growth is the best indicator of the level of resistance. Consider this section.

We removed this section in the revised manuscript.

Reviewer #2 (Remarks to the Author):

The study by Rogan et al presents new findings addressing the regulation of apoplastic proline levels with respect to PTI against *Pseudomonas syringae*. Plant-exuded proline serves as nutrient for *Pseudomonas* and the respective uptake from the apoplast via the plant transporters ProT2 and LHT1 might serve as tool contributing to plant induced immunity.

The study is very well designed, with a properly controlled setup of experiments. The results are trust-worthy. Besides, the manuscript is written in a very nice, coherent style. The results are of significance for the field of plant defense and connect to previous studies where amino acid transport was coupled to plant defense responses.

The only comment that I have deals about Figure 7 and the fairly quick turn to the amino acid transporter LHT1. Some more experiments would be needed to justify the drawn conclusions about the central involvement about the transporter:

1. Have the authors also checked the involvement of other transporters such as some of the AAPs, that can also contribute to proline transport? (line 334)

We have not tested other AAPs for their involvement in proline uptake. Both PROT2 and LHT1 were logical candidates for flg22-induced proline uptake based on previous evidence that they transport proline and are both transcriptionally induced by MAMPs. Because loss of *LHT1*

completely alleviates flg22-induced depletion of apoplastic proline, we have not continued to screen additional transporter mutants for this particular phenotype.

2. Also, it would be interesting to see the response in a *lht1*prot2* double mutant line. (line 335)

We agree that it would be interesting to test if loss of *LHT1* can counteract the increased levels of apoplastic proline that occur in *prot2* mutant leaves. However, because the focus of this work is flg22-induced removal of apoplastic proline, and loss of *LHT1* completely alleviates flg22-induced depletion of apoplastic proline, characterizing the phenotypes of an *lht1 prot2* double mutant is not necessary to support the main conclusions of this work.

3. The authors could support their hypothesis by looking at published phospho-proteomic data and check if LHT1 has been found being phosphorylated. (line 352)

We searched existing phosphoproteomics databases (PhosPhAt 4.0 and P3DB) and did not identify phosphopeptides derived from LHT1. Of course, because current phosphopeptide-enrichment and mass spectrometry protocols cannot identify all phosphopeptides present in a sample, we cannot make conclusions due to lack of evidence. A more targeted proteomics approach may detect LHT1 phosphorylation or other modifications, though these experiments are outside of the scope of work within this manuscript.

4. Did the authors check for regulation of Prot2 / LHT1 with respect to flg22-treatment?

We used RT-qPCR to measure the levels of *LHT1* transcripts in adult Col-0 leaves and confirmed that *LHT1* transcripts significantly increase 30 minutes after treatment with flg22. These data are now included as Fig S6a. We have not tested *PROT2* transcripts, though *PROT2* induction by flg22 treatment has been reported and we cite this study in our manuscript on line 276.

5. The authors cite that overexpression of LHT1 does not lead to decreased proline levels in the apoplast. It would be nice to include this line in the experimental setup as control. How is it for Prot2?

We agree that phenotypes of transgenic lines that overexpress *LHT1* and *PROT2* may be informative regarding how transporter activity may be regulated in MAMP-treated leaves. However, experiments with these lines, particularly plants that overexpress *LHT1*, are not necessary to support our conclusions in this manuscript and are better suited to future studies of how LHT1 transporter activity is regulated during defense.

Minor comment:

6. There are a lot of partially unnecessary abbreviations, e.g. line 105: MM and those abbreviations are not consequently followed, e.g. line 196, minimal media.

Thank you for catching this error. We corrected the MM abbreviation (now line 198 in revised manuscript). Within each figure legends we re-define MM as “minimal medium” for each first occurrence of the term, then “MM” for subsequent uses.

Besides, I have no further comments and I thank the authors for a very interesting study!

Reviewer #3 (Remarks to the Author):

The authors study the role of flg2 on PTI induction, for that they analyse the extracellular fluids of Arabidopsis leaves after elicitation with the flg2. In a metabolomics approach the authors have evidences that point out for the specific accumulation or depletion of several small compounds and test some for the hrpL-inducing activity. They focus then on proline and conduct several experiments to prove that proline presents a relevant role in the infection outcome. The authors also look into some proline transporters to access their role in the proline depletion seen and suggest that LHT1 may be a player on this regulation. Overall the approach is very interesting and important however in some cases it is difficult to follow the authors rational. I also believe that some of the conclusions that are drawn are not supported by the approach/results (see comments in the MS). The authors should assess these questions before the manuscript is considered for publication at Nature Communications.

line 7 - Needs to be better explained

Line 7 within the Introduction provides a general overview of immunity in animals. It is not clear whether the reviewer is asking here for more details about the animal immune system, or for revisions to improve the clarity of our writing. We have not modified this section in the revised manuscript, and we would appreciate more specific guidance about how to improve this section.

line 63 - the legend of figure 1 must be improved, A and B panels should have their own legends

Thank you, revised.

line 76 - accumulation instead of expression?

The *hrpL*_{promoter}:*gfp* is a transcriptional reporter and does not measure *hrpL* transcripts or HrpL protein levels directly. Therefore, in our opinion, expression is a better term than accumulation in this circumstance.

line 78 - accumulation instead of expression?

Please see above comment.

line 81 - I don't agree, the results confirm that either flg22 or elf26 decrease the AvrPto and accumulation and flg22 decrease the accumulation of T3SS proteins. Acannot ne generalized. (It cannot be generalized?)

Thank you. We revised our conclusions to be more specific: (line 81) “Together, these results confirm that flg22-and elf26-induced defenses restrict the expression of T3SS genes in *P. syringae*, and reveal that flg22-induced restriction of T3SS genes does not occur in QKO mutant leaves.”

line 93 - what is the estimated % of cytosolic contamination?

The average MDH activity values were 0.083 (mock-AWF), 0.082 (flg22-AWF) and 2.16 (total leaf). Based on these values, MDH levels in mock-AWF and flg22-AWF were 3.8% of the MDH levels in total leaf extracts. Note that in Figure 2b we revised “30-fold” to “25-fold” to more accurately describe this difference.

line 97 - The authors cannot state this based on the ref presented. In that study the authors shown that after elicitation the media (cell culture) suffered alkalization. On that study not apoplast alkalization was measured.

We now reference on line 98 an additional study that reports alkalization of the Arabidopsis leaf apoplast.

line 101 - Alkalinization may occur due to ROS accumulation, were ROS accumulation in the APF (AWF?) measure?

ROS are reactive and inherently unstable. Therefore, it is highly unlikely that any ROS would be present in our AWF samples after isolation, chloroform extraction and lyophilization steps. Nevertheless, to directly address this question, we used a standard luminol-horseradish peroxidase assay to measure ROS levels in AWF isolated from mock- and flg22-treated leaves.

The AWF samples used in this assay were isolated and chloroform-extracted as described in our methods. Little to no ROS was detected in either AWF sample, whereas ROS was clearly detected in flg22-treated leaf tissue included as a positive control.

Detection of ROS in apoplastic wash fluid (AWF). AWF was isolated from Col-0 leaves treated with mock or 100 nM flg22 for 8 hours. 20 μ L of each AWF sample was tested for ROS using a luminol-based horseradish peroxidase (HRP) assay. Graphed are luminescence readings 30 minutes after luminol/HRP addition, n = 3. Leaf tissue treated with mock- or 1 μ M flg22 for 30 minutes was included in the experiment as a positive control.

line 120 - figures 2D and 2F may be combined

Thank you for this suggestion. Our preference is to keep the figure panels in their current form, to preserve a logical order of figure panel call-outs within the results section.

line 127 - only the aqueous phase right?

Yes, we corrected this statement.

line 134 - only one of the fractions was analyzed. The aqueous phase where growth differences were seen?

Yes, correct. In the Methods section we added the following text to clarify what fraction was used for GC-MS: (line 640) “a 20 μ L aliquot of the aqueous phase from chloroform-extracted AWF”.

line 174 - why ribitol?

Ribitol was added to infiltration fluid as an internal standard for normalization purposes. Ribitol is useful as a normalization standard because a) it is not naturally present within Arabidopsis apoplastic wash fluid, and b) unlike proline, ribitol did not decrease in abundance over time when infiltrated into the apoplast (Fig. 4a).

line 177 - ¹³C proline levels decrease in the apoplast in control situation (without flg22), why?

This is a good question. We do not know the mechanism for why ¹³C-proline is rapidly depleted from the apoplast of mock-treated leaves. Most likely this is due to the activity of transporters that are active in the absence of flg22 treatment.

line 246 - I cannot follow the authors rationale on the negative regulation of apoplastic proline, to show that, in my opinion, the authors should analyse a line with none functional proT1 and 3, to prove that the levels of proline in the apoplast would decrease based on the activity of proT2. What the authors may prove is that proline accumulates in ProT2 mutants (probably due to the lack of transport to the cytoplasm) and that accumulation promotes bacterial growth.

In Fig S5a we show that apoplastic proline levels are increased in *prot2-3* but not in *prot1-1* or *prot3-2* mutant leaves. We confirmed this phenotype of *prot2* using the alternate allele *prot2-1* (Fig S5d).

line 250 - I cannot find evidence for the rapid induction of Prot2 in this reference

Expression data for *ProT2* can be found in Supplemental Table 1 of this reference. *ProT2* (AT3G55740) shows 12-fold induction one hour after flg22 treatment.

line 252 - I am also not understanding the rationale, based on the above experience the authors see that the levels of proline are only with the *prot2-3* mutants. Why using double mutants now?

The experiments in Fig 6f were done at a time when we were growing only the *prot2/prot3* double mutant instead of the *prot2* single mutant. Because the *prot3* single mutant did not have altered apoplastic proline levels (Fig S5a), the increased proline levels observed in the *prot2/prot3* double mutant (Fig 6f) can be attributed to the presence of the *prot2* mutant allele. Also, because we found that flg22-induced proline depletion was not altered in *prot2/prot3* (Fig 6f), repeating these experiments with the *prot2* single mutant is unlikely to change the outcome or conclusions of this experiment.

line 314 - flg22 instead of PAMP

Thank you, corrected.

REVIEWERS' COMMENTS

Reviewer #1 (Remarks to the Author):

The authors have satisfactorily addressed my concerns and also, in my opinion, those concerns of the other reviewers.

Reviewer #2 (Remarks to the Author):

Thank you for addressing my comments, and adding new data (gene expression data in Fig S6a) to the manuscript.

I have no further comments and recommend the study for publication.

Reviewer #3 (Remarks to the Author):

The authors have taken the reviewers comments and suggestions under consideration improving the manuscript. My recommendation is Accept.

minor points:

in the introduction, line 7- the revisions needed were to improve the clarity of writing.

regarding the alkalization, thank you for measuring the ROS levels but in fact with the protocol used for apoplastic fluid extraction, ROS might be absent. However the presence of ROS in the APF in biotic stress conditions was already reported and it is known that apoplastic ROS are produced by extracellular peroxidases and plasma membrane-bound NADPH oxidases, Rboh. Probably if a simpler APF extraction method was used (eg. cold water), ROS could be traced.

Reviewer #3 (Remarks to the Author):

minor points:

in the introduction, line 7- the revisions needed were to improve the clarity of writing.

Thank you, we have modified this sentence.

regarding the alkalization, thank you for measuring the ROS levels but in fact with the protocol used for apoplastic fluid extraction, ROS might be absent. However the presence of ROS in the APF in biotic stress conditions was already reported and it is known that apoplastic ROS are produced by extracellular peroxidases and plasma membrane-bound NADPH oxidases, Rboh. Probably if a simpler APF extraction method was used (eg. cold water), ROS could be traced.

Thank you for this explanation and suggestion.